# Shifting dominant periods in extreme climate impacts under global warming

Karim Zantout [1,2] ✉, Juraj Balkovic [3], Maik Billing[1], Christian Folberth [3], Simon N. Gosling [4], Tobias Hank [5], Stijn Hantson[6], Toshichika Iizumi [7], Akihiko Ito [8], Jonas Jägermeyr [1,9,10], Atul K. Jain [11], Nikolay Khabarov [12], Sian Kou-Giesbrecht [13], Fang Li [14], Mengxue Li [15,16,17], Tzu-Shun Lin [18], Wenfeng Liu [15,16,17], Christoph Müller [1], Masashi Okada [19], Sebastian Ostberg [1], Kedar Otta [19], Sam Rabin[18,20], Christopher P. O. Reyer [1], Clemens Scheer [20], Julia M. Schneider [5], Florian Zabel [21], Katja Frieler [1] & Jacob Schewe [1]

Spatio-temporal patterns of extreme climate events have been extensively studied, yet two questions remain underexplored: Do such events occur regularly, and how do regularity patterns change under global warming? We address these questions by investigating dominant periods in crop failure, heatwave, and wildfire data. Here, we show that under pre-industrial conditions dominant periods emerge in 28% of cropland exposed to crop failure and 10% of wildfire-affected areas, likely related to climatic oscillations such as the El Niño-Southern Oscillation, while heatwaves occur irregularly. The number of dominant periods increases by 2–13% during the transition from the pre-industrial era to the anthropocene. In the anthropocene, the occurrence of extreme events shifts towards monotonic growth, replacing previous natural regularity patterns. Linearly de-trended projections reveal an additional shift towards smaller dominant periods due to climate change. These shifts in regularity are crucial for adaptation planning, and our method offers an additional approach for studying extreme events.

Natural disasters from extreme climate events caused 2 million deaths, and 3.6 trillion US$ of economic loss between 1970 and 2019[1]. While risk reduction and early warning have significantly reduced fatalities over time, economic losses have increased by a factor of seven during this time period. Moreover, climate-related natural disasters caused 26 million new internal displacements in 2023[2]. These observations are connected to increased extreme climate events due to anthropogenic forcing and are expected to further aggravate under continued warming[3–7]. While there is large agreement on the increase in intensity and frequency of extreme climate conditions, only very few studies have investigated regularity of extreme event occurrence[8–12]. Yet, a better understanding of the temporal dynamics of extreme climate impacts is crucial for developing adaptation plans as regularity

ultimately relates to the predictability of extreme climate events. A large class of time series approaches has been deployed to determine the characteristics of extreme events[10], including extreme value theory[13,14] to determine return periods, spectral analysis[15,16] to extract periodic behavior, and stochastic or machine learning models[17,18] to understand underlying processes and make predictions. Within noisy extreme event time series, we define temporal regularity as variability that is essentially described by a single dominant period. To this end, classic Fourier analysis in combination with statistical tools allows us to determine whether a dominant period exists in noisy extreme event time series (see details in Section "Methods").

Here, we consider three types of extreme climate events, namely crop failures, heatwaves, and wildfires. While the definition of

heatwaves only depends on temperature, crop failures and wildfires are derived from impact models that combine the temporal evolution of climatic variables with soil and vegetation characteristics. Consequently, heatwaves correspond to a meteorological hazard whereas crop failures and wildfires are impacts. The inclusion of both allows us to investigate direct climate change effects (heatwaves) and the interplay with impact related dynamics (crop failures and wildfires). Wildfire studies have identified distinct fire domains that are connected to regrowth dynamics[19] and correlations between wildfires and the El Niño-Southern Oscillation (ENSO)[20,21]. For crop yield, only effects from large-scale climate oscillations on crop yield variability have been studied[22–27], such as ENSO, the Indian Ocean Dipole (IOD), and the North Atlantic Oscillation (NAO). However, it is unclear whether such regularities of crop yield also translate into similar patterns of crop failures. These studies on crop impacts and wildfire are based on historical observations and it remains unclear how these temporal features change under global warming. Only one recent study investigates the relationship between ENSO and crop impacts for future scenarios[28]. In the case of heatwaves, climatic oscillations have been shown to be strongly linked to the emergence of extreme heat[29–32]. Consequently, there is a strong relationship between climate oscillations and heatwave variability. In this study, however, we investigate whether extreme event variability can be described by a single dominant period and how this period evolves under climate change. In principle, the existence and value of the dominant period depends on the interplay of different climate modes and their interference with impact-type specific characteristics of the extreme event, e.g. land-use pattern, crop type and vegetation dynamics.

Our study is based on the latest climate model projections to drive an ensemble of climate impact models from the Inter-Sectoral Impact Model Intercomparison Project (ISIMIP). ISIMIP assembles harmonized inputs and provides consistent modeling protocols for a multi-model climate impact framework[33]. We use global gridded time series data from ISIMIP phase 3b to calculate crop failure, heatwave and wildfire time series from 1850 to 2100 for different scenarios. Note that also hydrological models take part in ISIMIP which would allow to define hydrological extreme events but due to the intricacies of the related model setups we restrict ourselves to agricultural and vegetation models and use heatwaves as a direct GCM derived event. Consequently, heatwaves are restricted to the hazard component as the full impact modeling, e.g. heat-related mortality, within ISIMIP is ongoing work. The extreme events are defined through percentiles of the respective pre-industrial extreme event indicator distribution (see Section "Methods" for details). The combination of climate and impact models allows us to investigate changes in dominant periods not only due to changes in the climate indicators but also their effects within impact models. Crop failure, heatwave, and wildfire are determined not only by the hazard itself but also by direct human influences affecting exposure and vulnerability which depend on socio-economic factors, e.g., resource scarcity, agricultural and forest management[34–37]. Such factors are difficult to disentangle and project into the future[38]. To investigate regularity patterns that exclusively arise from natural processes such as vegetation growth dynamics as well as temperature and precipitation variability, we here keep direct human influences such as land use changes and land management constant at 2015 conditions.

The dominant period is defined to be equal to the strongest periodic signal in the time series accounting for noise and its relation to other periodic signals in the time series (see Section "Methods" for more details). Consequently, the dominant period is only defined if the time series is sufficiently well described by this single periodic signal. This approach combines Fourier analysis with statistical tools and is motivated by the large size of the spatio-temporal data set within our multi-model setup, which calls for a simple characterization of regularity. For example, the detection of a dominant period of 2 years

indicates the prevalence of 2 years recurrence time in extreme event exposure. Both time series (1, 0, 1, 0, 1, 0, 1, 0) and (1, 1, 0, 0, 1, 0, 1) have average recurrence time 8/4 = 2 since we observe 4 events (1: event, 0: no event) in 8 time steps. On the other hand, the dominant period is 2 in the first (periodic) case and undefined in the second (irregular) case. The first case exhibits perfect periodicity which allows for more precise disaster management while the irregularity of the second case signifies another challenge in terms of risk expectation. This knowledge is relevant for insurances, disaster preparedness and response planning agencies, etc. In addition, regularity of extreme climate events may also help to identify critical thresholds for the recovery of affected systems, e.g. when extreme climate impact frequencies are larger than typical recovery times of ecosystems[39,40]. For example, in the case of wildfires, tree regeneration depends on the time period between severe fires and exhibits critical thresholds[41,42]. Consequently, the recovery risks for wildfires are smaller if wildfires exhibit regularity with a dominant period above the critical threshold. Note that dominant periods can be used in addition to average recurrence times to supply additional information on the regularity of time series.

Here, we show that dominant periods exist under pre-industrial climate conditions in 28% of cropland exposed to crop failure, 10% of grid cells exposed to extreme wildfire, and less than 1% in heatwave exposed grid cells. The observed dominant periods are between 7 and 13 years, likely related to climatic oscillations such as ENSO. During a transition phase 1950–1999 the number of dominant periods increases by 2%, 11%, and 13% for crop failure, heatwaves, and wildfires, respectively. This increase is related to higher event probabilities due to global warming. At the end of the 21st century, the extreme events exhibit non-linear monotonic growth which replaces previously observed regularity patterns. By removing the linear trend in the time series we are able to extract an additional effect from the strong warming, namely a general shift towards smaller dominant periods.

## Results
### Regularity under pre-industrial climate
The pre-industrial control (picontrol) setup within the ISIMIP framework[43] simulates stable pre-industrial climate conditions from 1850 to 2100 and serves as a reference for the future SSP scenarios (see Section "Methods").

The results are qualitatively independent of parameter choices for determining the dominant period while the multi-model, large time-scale setup allows for stable aggregated results (see Supplementary Discussion Sections 1.1 and 1.2). Additionally, we find that the dominant periods derived from the climate model-based impact simulations (ISIMIP 3b) for the historical period are consistent with those derived from impact models simulations forced by observational climate from ISIMIP3a[33] (see Supplementary Discussion Section 1.3). Therefore, we conclude that the climate models provide robust data for our subsequent analyses. Note that the presented results are medians over all climate impact model combinations and time windows that detect a dominant period. This approach is founded in technical limitations to detect dominant periods, e.g. spectral leakage, and different sensitivities of climate impact models to climate modes (see Section "Result aggregation").

As crop failure is defined with respect to the 2.5th percentile yield in the picontrol scenario, we can expect an average return period of 40 years for each crop type, maize, wheat, rice and soybean, and both irrigated and non-irrigated cropland separately which can assemble into higher frequencies depending on land use and the crop failure timings. For example, if all crop types fail in close-by years we can expect large dominant periods for the cropland-weighted exposure but if all crop types have equal land use share and fail equally distributed in time we can expect dominant periods down to $40/(4 \cdot 2) = 5$ years accounting for all four crops and both rainfed and irrigated

conditions. Calculating dominant periods for each crop type separately leads to the same dominant periods but with a different count and spatial distribution (see Supplementary Fig. 7).

We find evidence for regularity in the occurrence of crop failures in 28% of all affected cropland while 72% of the affected cropland show no regularity which corresponds to irregularity in 87% of all affected grid cells (gray color in Fig. 1a). The locations of regularity are in agreement with research linking crop yield variability to climate variability[22,23,25]. For example, the large cluster south of the Amazon coincides well with regions, where soy and maize have been shown to be significantly impacted by ENSO[22] and similarly for Eastern Africa, where wheat and maize yield variances are related to ENSO and the Indian Ocean Dipole (IOD)[25].

The calculated dominant periods are between 10 and 13 years (olive color) for large parts of agricultural areas, with periods of 7–10 years (ocher) also prevalent. A likely explanation for the dominant periods are influences from ENSO, IOD, and North Atlantic Oscillation (NAO)[24–26]. Note that NOA exhibits no clear low-frequency regularities but decadal variations[44] that may modulate dominant periods in the observed ranges. Similarly, IOD exhibits strong high-frequency modulations that cannot explain the observed dominant periods but IOD is correlated with ENSO[45,46] and may therefore influence the observed dominant periods. On the other hand, ENSO shows oscillations in the range of 2–8 years[47] and in addition to NOA and IOD is known to affect regional climate, and thereby also crops, across the globe[22–26]. Impacts on crops differ by crop type, region, and climate oscillation phase; e.g., some crops in some regions may be positively affected by a La Niña event and negatively affected or unaffected by an El Niño event, whereas for other crops and regions this may be different[26]. The observed periods are related to the aggregated crop failure resulting in an overlap of different frequencies, phases, amplitudes and land use patterns that add up to the calculated dominant period. Correlation analysis indeed shows near-zero correlation between aggregated crop failure affected area and the Southern Oscillation Index while crop specific analysis reveals non-zero model-median correlation in the case of maize, rice, and wheat (see Supplementary Figs. 8a and 9). Therefore, dominant periods of 7–13 years, i.e., longer than the typical period of ENSO, arise due to the interplay of different stochastic climate oscillations and sensitivities of the different crops contributing to the overall yield calculated for the grid cell. Moreover, other climate modes that exhibit decadal variation such as the Pacific Decadal Oscillation (PDO) and Atlantic Multidecadal Oscillation (AMO) have been shown to regionally affect crop yields[48,49] and therefore may locally influence dominant periods. In addition, the underlying strict definition of crop failure has an influence on the calculated dominant period. The 2.5th percentile as reference value for a drop in crop yield to be extreme results in few event occurrences under mild climate fluctuations. A test with randomly sampled data confirms that these results are no methodological artifacts (see Supplementary Discussion Section 4).

Heatwaves are directly calculated from the climate model outputs based on the 97.5th percentile threshold for the daily Heat Wave Magnitude Index (HWMId). We therefore expect more immediate effects from climate variability on heatwaves and therefore fewer occurrences of dominant periods as fluctuations are not mediated through vegetation or soil buffers like for crop failure and wildfire. We find that 99% of all affected areas exhibit no regularity (see Fig. 1b). Based on our extreme event definitions, heatwaves can be expected to occur approximately every $1/2.5\% = 40$ years which strongly limits the occurrence of dominant periods in the range 1–25 years. Exposure to heatwaves and therefore dominant periods appear with typical values for dominant periods between 7 and 13 years appearing in close to 1% of all affected areas. These rare occurences of regularity are consistent with influences of natural climate oscillations driving climate regularity on global scale. For example, we find a correlation coefficient of 0.37

between global heatwave affected area and the Southern Oscillation Index (see Supplementary Fig. 8b).

Wildfire dynamics involve several time scales related to climate oscillations and different vegetation growth dynamics[19,20]. We define extreme wildfire through a threshold on the annual burnt area given by the 97.5th percentile of the pre-industrial distribution (see Section "Methods"). We observe more irregularity in extreme wildfires than regularity with 90% of all impacted grid cells exhibiting no dominant periods. We observe dominant periods in the range of 4–13 years, similar to the ones for crop failures and heatwaves. These occurrences of dominant periods appear in all world regions in agreement with previous studies showing an influence of climate modes on wildfires[20,21]. The largest cluster is observed in South America where previous studies detected that El Niño 2015/16 led to the largest fire response[50]. The correlation coefficient between global extreme wildfire affected area and the Southern Oscillation Index is 0.22 (see Supplementary Fig. 8c) which is smaller than the respective value for heatwaves and consistent with wildfire dynamics being dependent not only on climatic factors but also on the biome types[19]. Our definition of extreme wildfire does not resolve the vegetation regrowth dynamics. To extract the influence of biome types and regrowth dynamics we applied a different classification of wildfires based on area thresholds in Supplementary Discussion Section 5 and find dominant periods that are consistent with observed wildfire dynamics.

## Climate change impact on regularity

The total area affected by extreme events in a moving 50 years window (see Eq. (3)) calculated for historical and future SSP5-8.5 simulations shows mostly constant total affected area up to the year 1950 from which point onward the area increases (see Fig. 2a–c). The stability of climatic pre-industrial conditions results in low variance of total affected area for all three event categories (see Fig. 2d–f).

For this reason we can describe the time window 1950–2000 as the transition period between the pre-industrial regime and the warming future while the period 2040–2069 will serve as future reference frame. This transition period is not observed in the respective affected area under pre-industrial climate conditions (see Fig. 2d–f). Note that this classification is only valid because we keep socio-economic conditions fixed while, for example, observed wildfires have drastically declined due to human activity in recent decades[35]. A more detailed picture of affected area counts for all grid cells is presented in Supplementary Discussion Section 6. We also present the results for the SSP1-2.6 and SSP3-7.0 scenarios in the Supplementary Discussion Section 7. For all event categories we observe qualitatively consistent results with the main difference being the weaker warming effect in SSP1-2.6 and SSP3-7.0 leading to smaller effects compared to SSP5–8.5.

Note that changes in median dominant periods may stem from a change in the set of climate impact models detecting a dominant period (see Supplementary Discussion Section 8). This change in the set of dominant period detecting models is due to different sensitivities of climate impact models to climate modes (see Section "Result aggregation") and challenges to detect regularity in noisy data within a single time window (see Supplementary Discussion Section 1.1).

For crop failure, the share of affected cropland without dominant period is 1.4% smaller for the transition period 1950–1999 than under pre-industrial conditions which corresponds to a decrease in affected grid area by 1.3% (inset in Fig. 3a). This increase in dominant periods appears at lowest dominant periods and at the cost of the previously strong dominant period signal at 10–13 years, where the time series patterns in 1950–1975 are not anymore congruent with the trend in 1975–2000 and the share within the dominant periods drops from 8.3% to 5.6%. This signals a shift through transitioning from previously stable pre-industrial climate conditions to a drastically warming world which is also visible in the reduced number of models

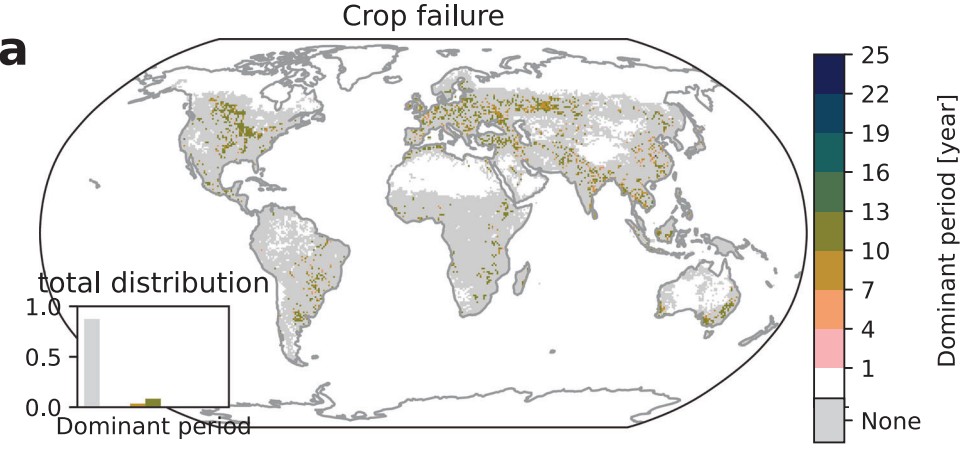

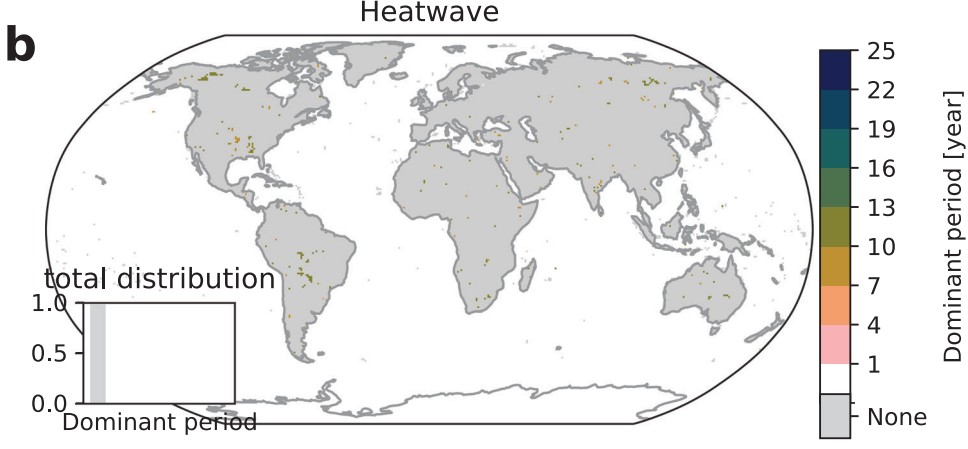

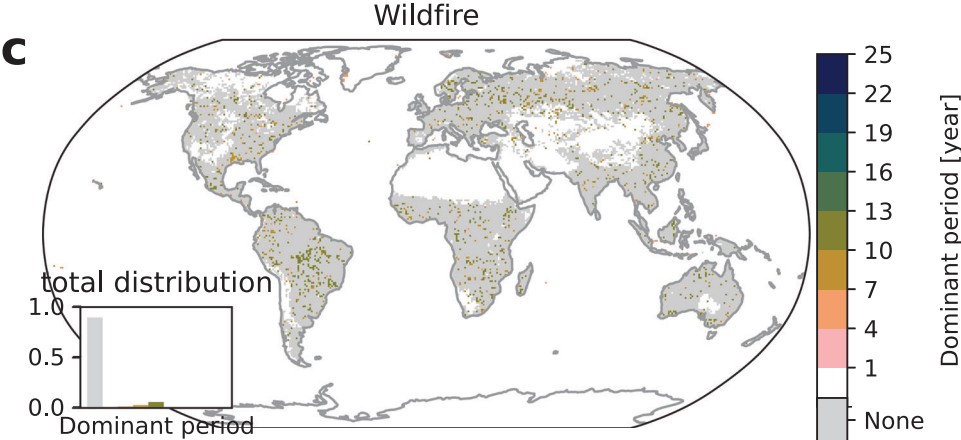

**Fig. 1 | Pre-industrial dominant periods.** Median dominant period for **a** crop failure, **b** heatwave, and **c** wildfire for picontrol aggregated over all time windows 1850–1899, 1900–1949, …, 2050–2099 and climate-impact models. The white color signifies no extreme climate impact occurrence and gray color signifies no dominant period (irregularity) while existing dominant periods are grouped in three-year regularity intervals ranging from 1–4 years (pink) to 22–25 years (blue). The inset shows the distribution of the dominant period counts.

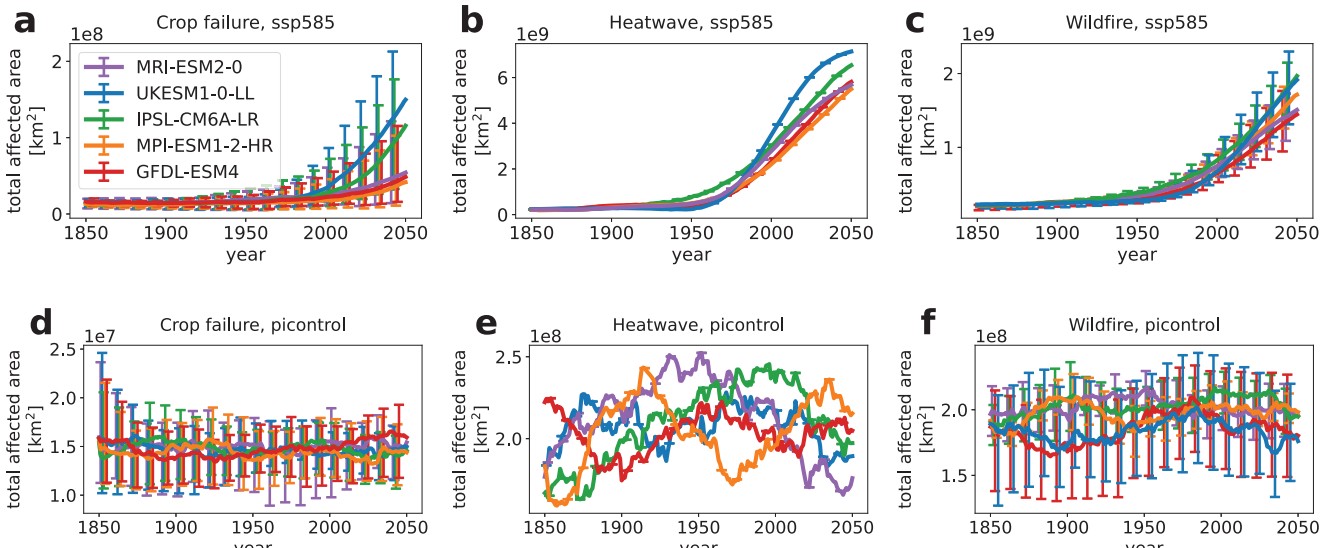

**Fig. 2 | Extreme event affected area.** Total global area affected by **a**, **d** crop failure, **b**, **e** heatwave, and **c**, **f** wildfire accumulated within a moving 50 years time interval. The top panel **a**–**c** shows the SSP5-8.5 scenario while the bottom panel **d**–**f** shows the picontrol scenario. The colored lines indicate the median value in units of km² over all impact models for each GCM. The error bars show the range along different impact models (only applicable for crop failure and wildfire), where we show only every tenth bar for visibility. Note that all events occur on different scales.

that detect regularity (see discussion in Supplementary Discussion Section 8).

In addition, we observe an emergence of largest dominant periods between 22 and 25 years sparsely distributed in many world regions, e.g., South East Asia. The changing climate conditions that lead to a significant monotonic trend in the impact time series may result in 22–25 years dominant period (compare Fig. 7c). Within the transition phase we also identify shifts from previous regularity patterns towards highest dominant frequencies, namely an increase of 1–7 years dominant period shares in affected areas from 4% to 6%. These high frequency patterns are sparsely distributed and occasionally form small clusters in some regions, e.g., North America and South-East Asia. We attribute the transition towards smaller dominant periods at the cost of larger dominant periods under picontrol to the climate forcing trend (compare Fig. 2a), where irregularity can arise from strong climate forcing trends that interfere with quasi-periodic climate variability. The increase in higher frequencies may be related to stronger climate oscillations under strong global warming[51] which would shift the spectrum further towards the 2–7 years range for the prevailing ENSO contribution on crop yields[25].

This transition becomes more evident in 2040–2069 (Fig. 3b) where most world regions show only large dominant periods or no regularity. Note that we are considering a smaller time window at the end of the century to avoid the strong monotonic warming trend in extreme events which mainly determines the results in 2050–2099 (see Supplementary Discussion 9). The largest detectable dominant period in the time window 2040–2069 is 15 years but the largest dominant periods of 22–25 years are observed when the time window 2050–2099 is considered (see Supplementary Fig. 29a). This observation supports our interpretation of a transitioning phase where dominant periods change due to a shift from stable climatic conditions towards a new era in the anthropocene. Fig. 2a shows an increase in the range of total affected area from 1950 on. This general climate forcing trend supersedes possible climate regularity patterns that are smaller in magnitude and we therefore observe either no regularity or the global warming trend itself.

In order to understand whether pre-existing regularities persist underneath the global warming trend, we perform a linear detrending on the time correlation function (see Eq. (1)) and calculate dominant periods on this new time series. Linear detrending consists of an ordinary least squares regression for each univariate correlation function in each time window (see Eq. (1)) and subtracting the slope component from the time correlation function. While the trend in Fig. 2a is superlinear we still adhere to linear detrending as higher order detrending can remove periodicity signals when trend functions are nonlinear. The corresponding median dominant periods for 2040–2069 are shown in Fig. 3c. Removing the trend in the correlation function leads to a relative increase in irregularity and to a relative decline in largest dominant periods, which is consistent with the assumption that large dominant periods result from the global warming trend that can be mostly absorbed through detrending. Moreover, we observe a minor resurgence of regularity signals with high-frequency events increasing in comparison to lower frequencies. For example, dominant periods in the range of 1–4 years are more frequent than those with 4–7 and 7–10 years, which is a significant shift compared to pre-industrial results (Fig. 1a). This further shift to lowest dominant periods is consistent with our previously mentioned explanation of increasing ENSO intensity dominating over previously existing regularity patterns. Note that by considering $2\Delta T = 50y$ we find more irregularity in the time window 2050–2099 due to the super-linear warming effect that cannot be sufficiently absorbed with linear detrending (see Supplementary Fig. 29). We therefore conclude that crop failure regularity is shifting from a pre-industrial distribution to a transition phase with declining 10–13 years dominant periods and subsequently to a warming future where the superlinear climate forcing is characterized by both irregularity and a strong trend with suppressed natural regularity signals.

In contrast to the crop failure results we observe the emergence of regularity in extreme heatwaves for the historical interval 1950–1999 (see Fig. 4a), namely from below 1% to 10% in the total distribution compared to the picontrol results.

Here again the regime shift around 1950 is obvious in the total affected area (Fig. 2b) which leads to the emergence of regularity in the transitioning phase. Regularity occurs since the probability for heatwaves increases with global warming while the heatwave expectation value under picontrol was one extreme event in 40 years. Increases in temperature variability due to human forcing were already observed in similar modeling setups[52] and support our interpretation of the results

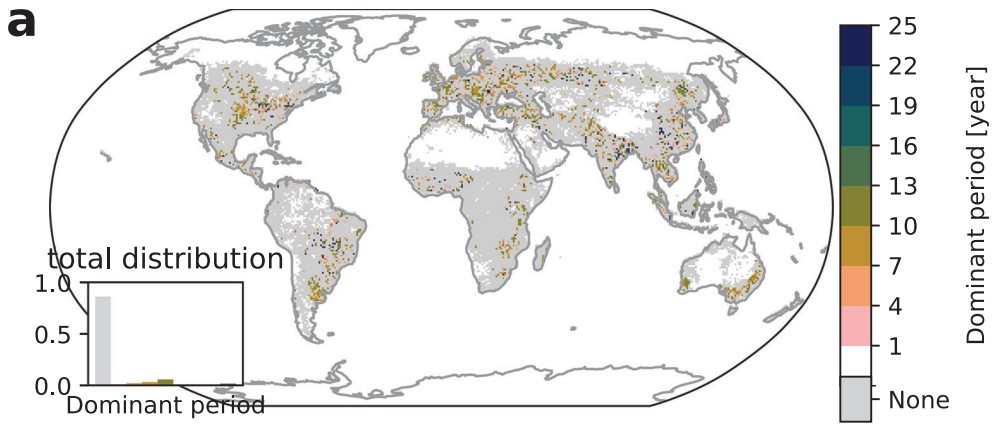

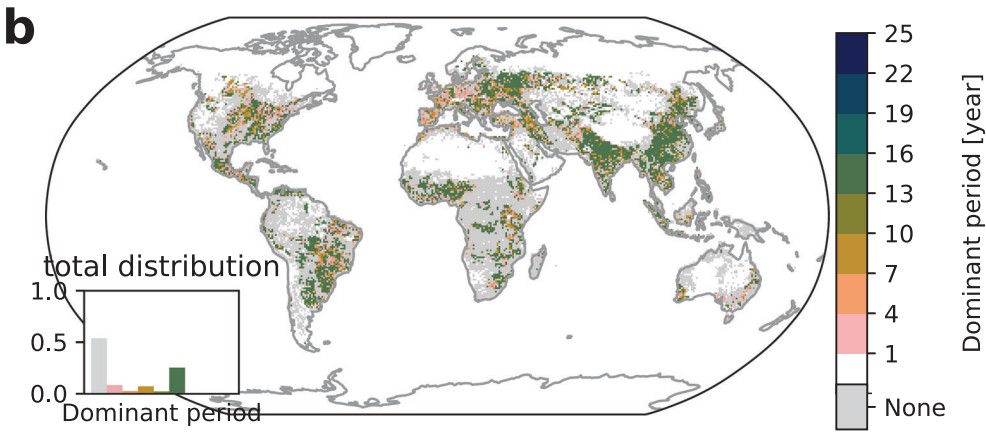

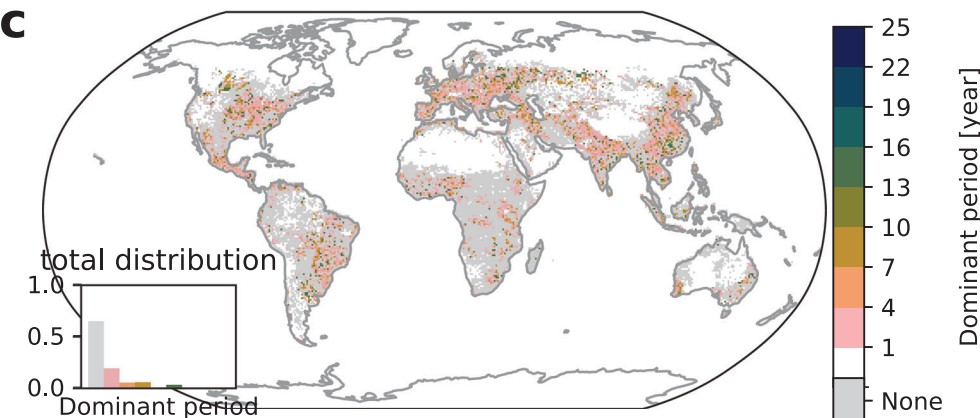

**Fig. 3 | Crop failure dominant periods for SSP5–8.5.** Median dominant period of crop failures for RCP8.5 and the periods **a** 1950–1999, **b** 2040-2069, and **c** linearly detrended 2040–2069. The white color signifies no extreme climate impact occurrence and gray color signifies no dominant period (irregularity) while existing dominant periods are grouped in three-year intervals ranging from 1–4 years up to 22–25 years. The inset shows the distribution of the dominant period counts.

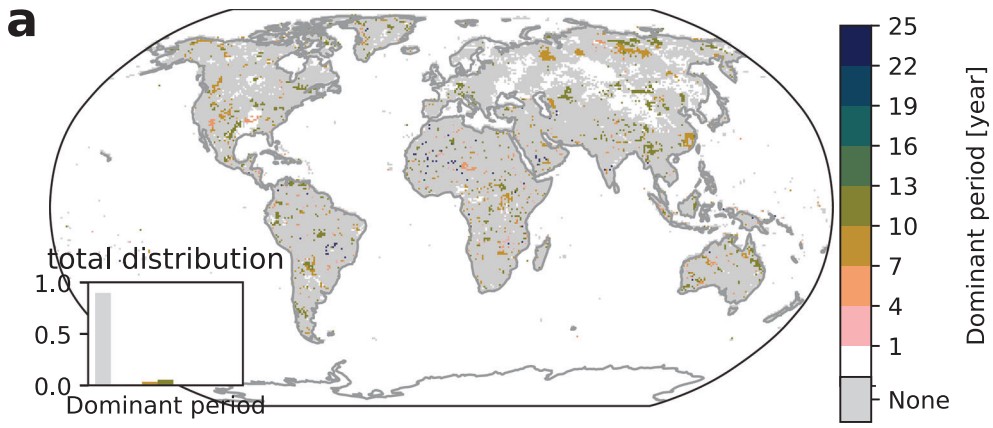

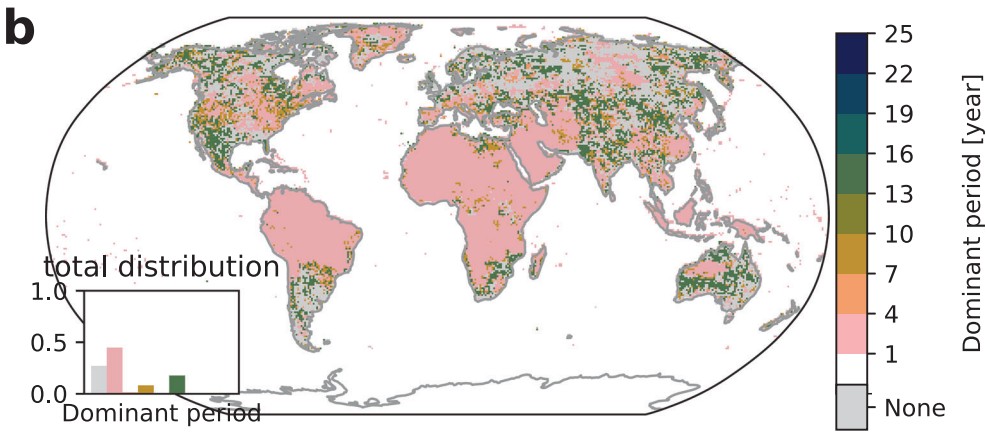

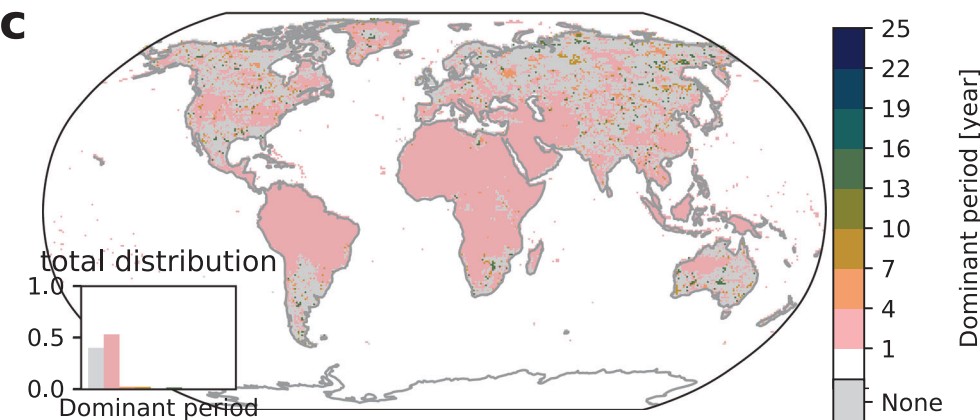

**Fig. 4 | Heatwave dominant periods for SSP5–8.5.** Median dominant period of heatwaves for **a** 1950–1999, **b** 2040–2069, and **c** linearly detrended 2040–2069. The white color signifies no extreme climate impact occurrence and gray color signifies no dominant period (irregularity) while existing dominant periods are grouped in three-year intervals ranging from 1–4 years up to 22–25 years. The inset shows the distribution of the dominant period counts.

in the transition phase. The emergent regularity patterns are similarly distributed to the crop failure dominant periods with highest occurrences of dominant periods in the range of 10–13 years and we find heatwave clusters in all world regions.

This picture changes again drastically in the 2040–2069 period where dominant periods can be determined in almost all regions with a clear signal from dominant periods of 1–4 and 13–16 years or even more years (see Supplementary Fig. 30a). The regularities from 1 to 4 years are saturated values where the picontrol threshold values is exceeded almost every year (see Supplementary Fig. 23b) while largest dominant periods speak of a clear warming trend that dominate the time series. Highest frequencies at low latitudes are consistent with previous studies of future scenarios on heat indicators[53,54].

After linear detrending (see Fig. 4c) we find a resurgence of irregularity (40% in the total distribution) and prevailing highest frequency clusters between 1 and 4 years (pink) across all world regions. This increase towards smallest dominant periods may be not only related to the general increase in temperature within SSP5–8.5 but also to influences from stronger ENSO effects with a fingerprint in the 2–7 years range.

In the case of wildfires, we observe a decrease in irregularity from 90% under pre-industrial conditions to 77% in 1950–1999 in the total distribution of dominant periods (see Fig. 5a).

This change is accompanied by an increase of regularity in the 4–13 years frequency regime from 10% to 22% in total affected area that appears in all world regions. The source of this shift is again the stark regime shift around 1950 (see Fig. 2c) which leads to an increased impact probability. Our findings are consistent with observations of increased frequency and intensity of extreme wildfires due to climate change[55].

Moving to the 2040–2069 period (Fig. 5b) we find extensions of the shortest dominant period regions in Africa, both Americas, and Asia. In addition, we again find longest dominant periods, e.g. in the Amazon and Southern Africa, which may hint that the previously observed dominant periods are now replaced by the pure warming trend which would lead to longest dominant periods. We find that wildfire regularity patterns are more robust to global warming in comparison to crop failure and heatwaves which can be attributed to strong buffering effects of vegetation regrowth.

Linear detrending leads to an increase in irregularities at the cost of the largest dominant periods (see Fig. 5c) resulting in the highest count of dominant periods to occur at smallest dominant periods between 1 and 4 years affecting all world regions. Note that the dominant period of 10–13 years still remains significant and occurs in 4% of all dominant periods.

## Discussion

We have investigated the global distribution and dominant periods of three types of extreme climate events, namely crop failure, heatwave, and wildfire.

We find that 28% of the cropland exposed to crop failures worldwide and 10% of grid cells exposed to wildfires are regular under pre-industrial climate conditions with dominant periods concentrated at 10–13 years. Such dominant periods are related to climatic oscillations such as ENSO, IOD, NAO which have been already shown to have a significant influence on crop yield variability[22,23,25,26]. Similarly, studies have found connections between climatic oscillations such as ENSO and wildfires[20,21] and also heatwaves[32]. These regularity patterns are concentrated in specific world regions, e.g., South Asia and South-East Asia, and Europe. The global warming trend within SSP5–8.5 replaces them almost entirely after a transitioning phase characterized by increased shift towards smaller dominant periods. Through linear detrending in time we were able to observe an additional shift towards higher frequency extreme events driven by the strong warming effect.

In summary, our analysis shows that existing natural regularity patterns not only depend on regularities in climate forcing but are also influenced by internal dynamics of the considered system itself. In addition, regularity patterns undergo a substantial shift through anthropogenic forcing in the transitioning phase from quasi-stable pre-industrial climate conditions towards the anthropocene, which leads to reduced dominant periods of crop failures and wildfires while extreme heatwave regularity emerges due to increased impact probability. This implies a reduction in predictability of crop failure and wildfire as periodic patterns in the extreme event occurrences change while rare heatwaves become more frequent. Previously significant regularity patterns that are related to climatic oscillations are overshadowed by signals of the anthropogenic warming trend and a shift to higher frequencies in extreme climate event patterns.

These findings not only further underscore the necessity for climate mitigation efforts but also imply that resource allocation for adaptation strategies have to account for decreased predictability of extreme event occurrences due to shifting dominant periods and the general warming trend. Insurances and disaster preparedness efforts are especially concerned and our regularity pattern and exposure maps may provide a useful tool to further assess regional strategies for the future.

## Methods

### Extreme weather event data
We investigate three types of extreme event, namely crop failure, heatwave, and wildfire. The exposure to crop failure and wildfire relies on process-based crop and vegetation model simulation results from ISIMIP 3b[33,43] that are driven by the bias-adjusted[56,57] daily output of Global Climate Models (GCMs) from Phase 6 of the Coupled Model Intercomparison Project (CMIP6)[58,59] (see Table 1). In contrast, we deduce the heatwave exposure directly from the bias-adjusted GCM output. The GCM and impact simulations are run for a historical (1850–2014) and a future period (2015–2100), where the latter is represented through the Shared Socio-economic Pathways SSP1–2.6, SSP3–7.0 and SSP5–8.5[59–61]. Additionally, we include a baseline simulation called picontrol with stable pre-industrial climate conditions for the full period (1850–2100). The GCM simulations are downscaled to a spatial resolution of 0.5° latitude and 0.5° longitude and the crop and wildfire model simulations have the same resolution.

To separate the effects of physical and biogeochemical processes from socio-economic dynamics in our analysis we fix the socio-economic conditions in all impact models to the ones in 2015. Crop failure calculations are based on maize, rice, soybean, and wheat yield estimates from 8 global gridded crop models (GGCMs; see Table 1), provided by AgMIP's Global Gridded Crop Model Intercomparison[62]. Note that some models distinguish between different wheat and rice types which we also preserve in our analysis. The crop models follow a harmonized simulation protocol with fixed agricultural management and land use assumptions around the year 2015. For further details, see ref. 62. Wildfire exposure is extracted from the output of the global terrestrial biosphere models CLASSIC, LPJmL-5-7-10-fire, and VISIT (see Table 1).

We define the extreme event exposure with respect to exposed area on an annual time scale based on ref. 3. While extreme climate impacts typically occur at smaller time scales we aggregate the exposure to a yearly time scale for better intercomparison and to investigate regularities beyond seasonality. For each grid cell, crop type (maize, rice, soybean, and wheat), and irrigation type (irrigated and rainfed) we define a crop to fail if the crop yield falls below the 2.5th percentile of the respective picontrol yield distribution. This threshold represents extreme events as such low crop yields occur on average once in 40 years. The exposed grid cell area fraction is set equal to the grid cell area fraction whereupon the respective failed crop is grown.

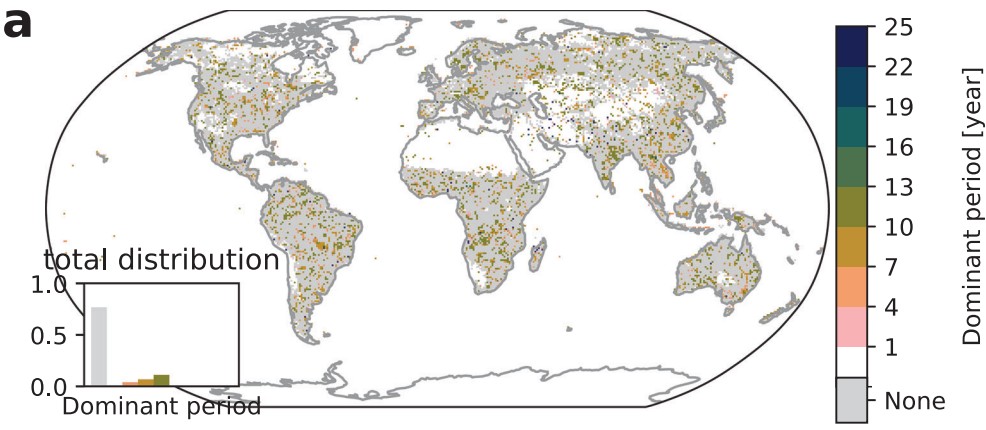

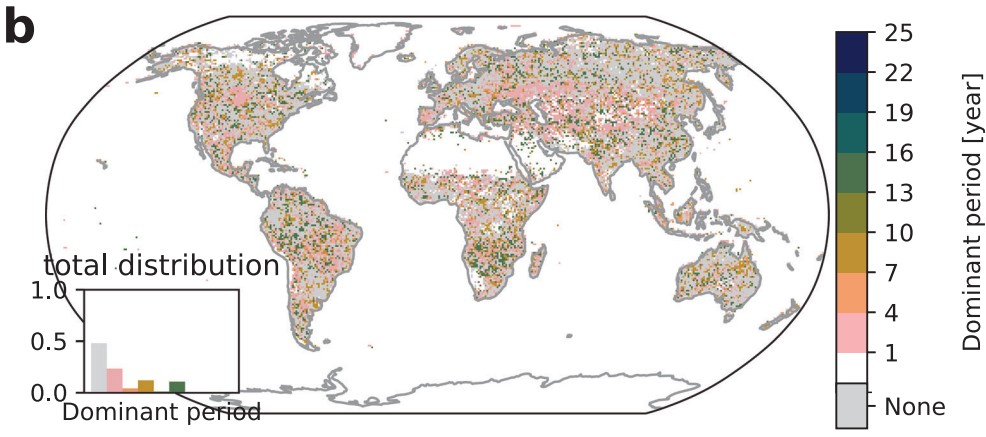

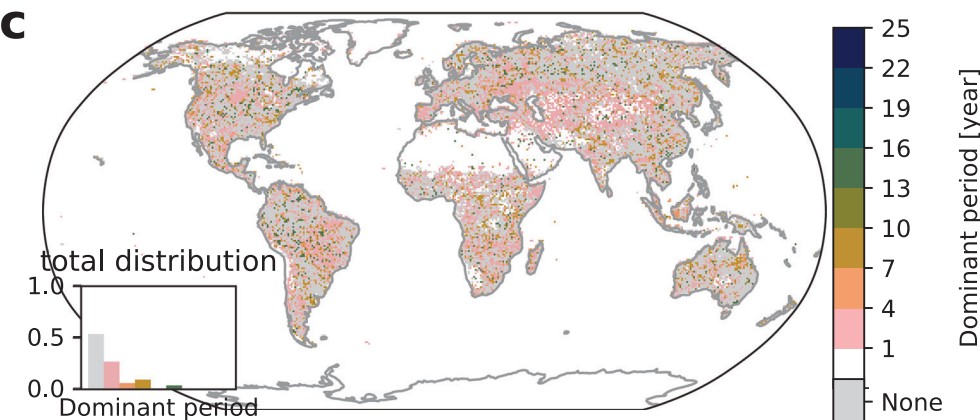

**Fig. 5 | Wildfire dominant periods for SSP5-8.5.** Median dominant period in wildfires for **a** 1950–1999, **b** 2040–2069, and **c** linearly detrended 2040–2069. The white color signifies no extreme climate impact occurrence and gray color signifies no dominant period (irregularity) while existing dominant periods are grouped in three-year intervals ranging from 1–4 years to 22–25 years. The inset shows the distribution of the dominant period counts.

**Table 1 | ISIMIP3b climate and impact models**

| | Impact model | Global climate model |
|---|---|---|
| crop failure | CROVER, CYGMA1p74, EPIC-IIASA, ISAM, LDNDC, LPJmL, PEPIC, PROMET | GFDL-ESM4, UKESM1-O-LL, IPSL-CM6A-LR, MPI-ESM1-2-HR, MRI-ESM2-O |
| heatwave | HWMId | GFDL-ESM4, UKESM1-O-LL, IPSL-CM6A-LR, MPI-ESM1-2-HR, MRI-ESM2-O |
| wildfire | CLASSIC[a], LPJmL-5-7-10-fire, VISIT | GFDL-ESM4, UKESM1-O-LL, IPSL-CM6A-LR, MPI-ESM1-2-HR, MRI-ESM2-O |

[a]Note that CLASSIC simulations are only available for GFDL-ESM4 and UKESM1-O-LL input.

The total exposed area is the aggregated value for all crop and irrigation types.

Wildfires are processes that exhibit strong geographical differences in terms of burnt area. In general, wildfires can lead to a continuous scale of burnt area ranging from 0 to 100% of a grid cell. Individual wildfires involve return periods between a few years and centuries. These stark differences in temporal scale are mainly driven by the vegetation regrowth dynamics, which differ between different biomes[19,20]. Taking the size of grid cells into account the burnt area at the grid-cell level is often the result of multiple wildfires occurring within that cell, affected by multiple overlapping regrowth cycles. Small burnt area fractions of a grid cell have little effect on the available fuel in the cell. Therefore, larger burnt areas are not inhibited in the following year; however, with a large fraction of the grid cell burnt, fuel recovery will take longer. Besides fuel availability, other factors such as fuel flammability, ignition sources and fire spread rate also vary over time and affect burnt area. Consequently, it is difficult to identify a single dominant period in the burnt area of these multi-scale fire systems. To disentangle the wildfire dynamics we apply two wildfire definitions. We define extreme wildfire based on the 97.5th percentile of the annual burnt area distribution under pre-industrial conditions in the main text. The exposed grid cell area fraction is one if the percentile threshold is exceeded and zero else. The analysis of wildfire emergence is presented in Supplementary Discussion 5 based on an upper exposed area threshold.

Finally, we calculate heatwaves from near-surface air temperature based on the Heat Wave Magnitude Index daily (HWMId)[63,64]. Specifically, we define a grid cell to be exposed to a heatwave if the HWMId of that year exceeds the 97.5th percentile of the picontrol distribution of that grid cell. Note that the impacts of heatwaves depend on the applied definition and that heatwave definitions may also be based on marine systems[65,66].

## Methodology

Given an extreme climate impact $i$, $i$ = cropfailure, heatwave, wildfire, we denote the affected area fraction in the grid cell $(\theta, \phi)$ and in year $y$ by $f_i^{jk}(y, \theta, \phi)$, where $j$ and $k$ denote the driving climate model and impact model, respectively. As the analysis is performed for each impact type, model combination, and grid cell position separately, we simplify the notation by dropping the respective indices. In order to investigate changing extreme event regularities we split the simulation interval 1850–2100 into five discrete intervals of lengths $2\Delta T = 50$ years. By increasing the window size to $2\Delta T = 60y$ we find no qualitative difference except for the reduced signal due to stronger decorrelation within the longer time window (see Supplementary Discussion Section 1.1). The window size $2\Delta T$ is chosen to (i) encompass sufficient data points to resolve global climate periodicity influences from El-Niño Southern Oscillation (ENSO), Indian Ocean Dipole (IOD), and North Atlantic Oscillation (NAO) regularities while (ii) limiting the window size accounts for temporal decorrelation. In the strong warming scenario we additionally consider $2\Delta T = 30$ years to account for the strong warming trend. We find no regularities on longer time scales when considering time windows of 250 years (see Supplementary Discussion Section 10).

For each time interval with lower boundary $t_0$, e.g. $t_0 = 1850, 1900, 1950, 2000, 2050$, we define the time auto-correlation function

$$C_{t_0 \Delta T}(n) = \frac{1}{\Delta T} \sum_{m=0}^{\Delta T - 1} f(t_0 + m)f(t_0 + m + n). \quad (1)$$

Accordingly, the auto-correlation function $C_{t_0 \Delta T}(n)$ contains information on the regularity of extreme event patterns within each time interval. If the impact time series were periodic, we can apply a discrete Fourier transformation to obtain the Fourier coefficients and learn about the extreme event regularities. Note that more advanced approaches[10,16,67,68] such as the Multi Taper method allows to minimize spectral leakage and reduce variance at optimized spectral resolution while autoregressive models can explicitly account for red noise. On the other hand, these elaborate approaches typically involve higher computational costs and hyperparameters that need to be estimated for each impact, scenario, model, or even region/location. In addition, the resulting power spectrum of such approaches is not discrete but continuous which makes it difficult to determine a single dominant period based on peaks and their relationship in terms of harmonics. Furthermore, statistical approaches in time series analysis[10,69] typically involve assumptions on the underlying process that are critical for the return period estimate. For example, Poisson processes are based on (heterogenous) rates that may depend on time, location, model etc. Within the ISIMIP setup this parameter is difficult to estimate as impact time series are provided as single realizations. Therefore, additional assumptions or approximations are needed to get enough data for fitting to a Poisson process. Estimates of return periods from probability distributions face these challenges from restricted time length in the case of wildfires[19] and in the case of generalized extreme value distributions of floods[70,71].

For our purposes we can rely on a simpler quantification of regularity as we are only interested in the dominant spectral feature. Additionally, we use the median and standard deviation from the multi-model distribution of dominant features to partly account for known caveats such as spectral leakage (see Section "Result aggregation"). As the climate event data is based on complex dynamical system simulations within CMIP6[58,59] we can only expect quasi-periodic signals in the impact time series. Regularities in the climate impact affected area may arise since quasi-periodic climatic conditions inhibit or boost extreme climate events. In addition, applied extreme event definitions limit the number of observations within the simulation data and therefore shape the regularity patterns that we can expect.

In order to separate the non-periodic signals within the extreme event data we apply the following scheme to determine a so-called dominant period.

- If the maximum of the auto-correlation function $C_{t_0 \Delta T}(n)$ (see Eq. (1)) is smaller than $\epsilon_0 = 10^{-4}$ we say that there is no dominant period, because the affected area is smaller than 1%. Else, continue with the next step.
- Apply a discrete Fourier transformation to the auto-correlation function $C_{t_0 \Delta T}$ to identify the Fourier coefficients $\{c_n\}_{n=0,...,\Delta T - 1}$.
- Sort the Fourier coefficients $\{c_n\}_{n=0,...,\Delta T - 1}$ in decreasing order $\{c'_n\}_{n=0,...,\Delta T - 1}$, namely $|c'_0| \geq |c'_1| \geq \ldots \geq |c'_{\Delta T - 1}|$.
- Determine $n \in \{0, 1, ..., \Delta T - 1\}$ such that the Fourier coefficients $c'_0, c'_1 \ldots, c'_n$ form a frequency sorted list containing a fundamental

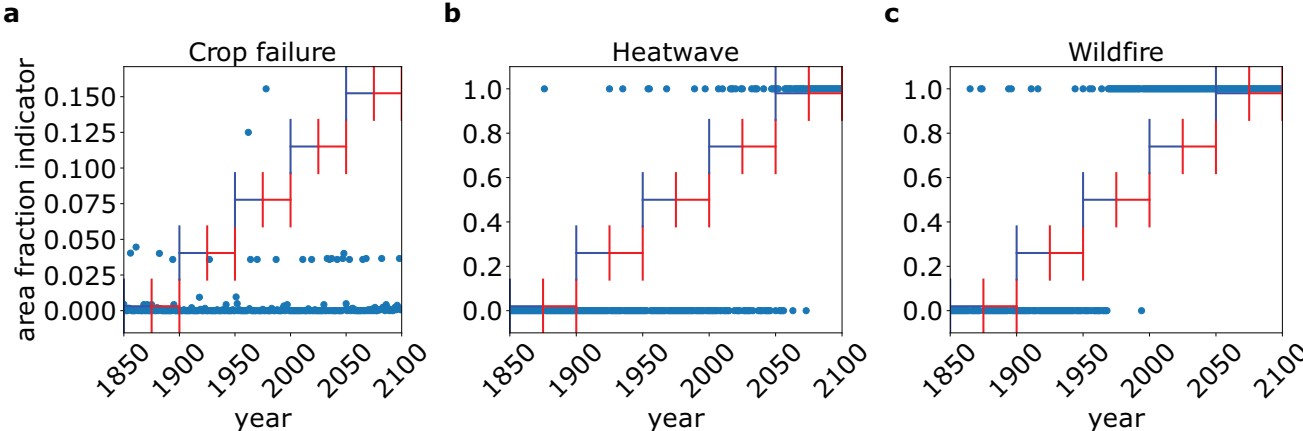

**Fig. 6 | Extreme event time series.** Area fraction affected by extreme events as a function of time. The blue and red whiskers denote the time windows 1850–1899, …, 2050–2099 that form the basis for the auto-correlation function in Eq. (1). More specifically, the blue section signifies the interval $(t_0, t_0 + \Delta T)$ while the red section marks $(t_0 + \Delta T, t_0 + 2\Delta T)$. The whiskers are shifted to avoid overlapping of red and blue segments. **a** Crop failure time series for climate model MPI-ESM1-2- HR, impact model EPIC-IIASA, for SSP5-8.5 at latitude longitude coordinates (−15.25, 28.25). **b** Heatwave time series for climate model GFDL-ESM4, impact model HWMID, for SSP5-8.5 at grid location (23.25, 72.25). **c** Wildfire time series for climate model GFDL-ESM4, impact model LPJmL-5-7-10-fire, for SSP5-8.5 at grid location (−15.25, 28.25).

frequency and higher harmonics. The list proceeds from lower to higher frequencies, where the constant signal $c_0$ can appear at any position. In the special case where the fundamental frequency is $1/\Delta T$ we take the frequency corresponding to the next largest Fourier coefficient.

- If the Fourier fit constructed with Fourier coefficients $\{c'_0, \ldots, c'_n\}$ explains at the majority of the time variation in $C_{t_0 \Delta T}$, namely the adjusted coefficient of determination $R^2$ fulfills $R^2 > 50\%$, we define the dominant period candidate as per$(c'_0)$ if $c'_0 \neq c_0$ and per$(c'_1)$ else. per$(c)$ denotes the period corresponding to the Fourier coefficient $c$.

- For the dominant period candidate per$(c'_0)$ we test if the respective peak $|c'_0|^2$ in the power spectrum is due to red noise[72,73] with a significance level of 95%. The red noise spectrum $|c_n|^2$ within an AR(1) model is distributed as

$$|c_n|^2 \sim \frac{1 - \phi^2}{1 + \phi^2 - 2\phi \cos\frac{2\pi n}{\Delta T}} \chi_2^2, \qquad (2)$$

where $\phi$ is the lag-1 correlation function and $\chi_2^2$ is the chi-squared distribution with 2 degrees of freedom.

We find that the choice of $R^2$ only enhances/suppresses the number of dominant periods when being decreased/increased but does not influence their values (see Supplementary Discussion Section 1.2).

While step 1–4 are essentially a Fourier decomposition where we define the dominant period through the largest non-trivial coefficient, we keep higher harmonics in the Fourier fit in step 5. The motivation for this approach is that even if underlying quasi-periodic natural processes influencing climate extreme event $i$ occur at high frequency we do not expect to observe the same periodicity in the impact signal due to thresholds in the extreme event definition and the stochasticity in the climate model input. For example, mild ENSO in some years may be insufficient to trigger an extreme climate impact which results in larger dominant periods even though the ENSO frequency may be a subdominant signal in the extreme climate impact time series. Allowing for higher harmonics in step 5 acknowledges for these effects by incorporating possible connections to faster natural processes. The $R^2$ criterion in step 5 accounts for the amount of variability explained by the dominant period. If the criterion is fulfilled we can state that at least

half of the variability is explained by the dominant period and its higher harmonics. In this sense the criterion makes sure that the dominant period is indeed significant and additionally accounts for stochasticity of the time series. As geophysical time series typically contain red and white noise[74,75] we perform a significance test in step 6 to make sure that the detected dominant periods are not noise artifacts[72,73].

In Fig. 6 we show extreme event time series for exemplary locations and model combinations.

The fluctuations within the data are most prominent for the non-extensive extreme event, namely crop failure, where the affected area can be smaller than the whole grid cell. The exemplary locations also show that the climate extreme event signals are themselves not obviously regular which supports our approach to look at periodicity not in the affected area but in the auto-correlation function (Eq. (1)).

The respective local auto-correlation functions for $t_0 = 1950$ are shown in Fig. 7.

The colored lines represent Fourier fits with the $n$ largest coefficients, $n = 1, 2, 3, 4$ according to step 1 to 4 in the definition of the dominant period. Corresponding to step 4 we show the Fourier fit that contains only a base frequency and higher harmonics in violet. In the case of crop failure (Fig. 7a) we find a dominant period of 8.33 years, where the base signal (yellow line) is connected to prominent peaks at time lag 0, 8 and 16 years. In addition, we find higher harmonics with smaller amplitudes that increase the amplitude at the previously mentioned time lags but also resolve smaller peaks, e.g., at time lag 2 and 14 years. With an adjusted $R^2$ value of 0.52 this dominant period lies at the lower bound of acceptance for regularity. For the heatwave example (Fig. 7b) we find no dominant period as we observe signals only at 0, 14 and 21 years time lag which is related to the underlying binary data which necessitates non-harmonic frequencies to reproduce the zero sections of the time series. As a result we have an adjusted $R^2 = 0.46$ and therefore we find no dominant period in this case. Finally, we detect a dominant period of 25 years for the wildfire example (Fig. 7c) that is related to the general increasing trend in the auto-correlation function. A discussion on the robustness of the method can be found in the Supplementary Discussion Section 1. A comparison of dominant periods in Fig. 7 and dominant periods estimated from the largest spectral peak of autoregressive models shows differences due to the additional steps 4–6 in our methodology and differences in the spectral estimates themselves

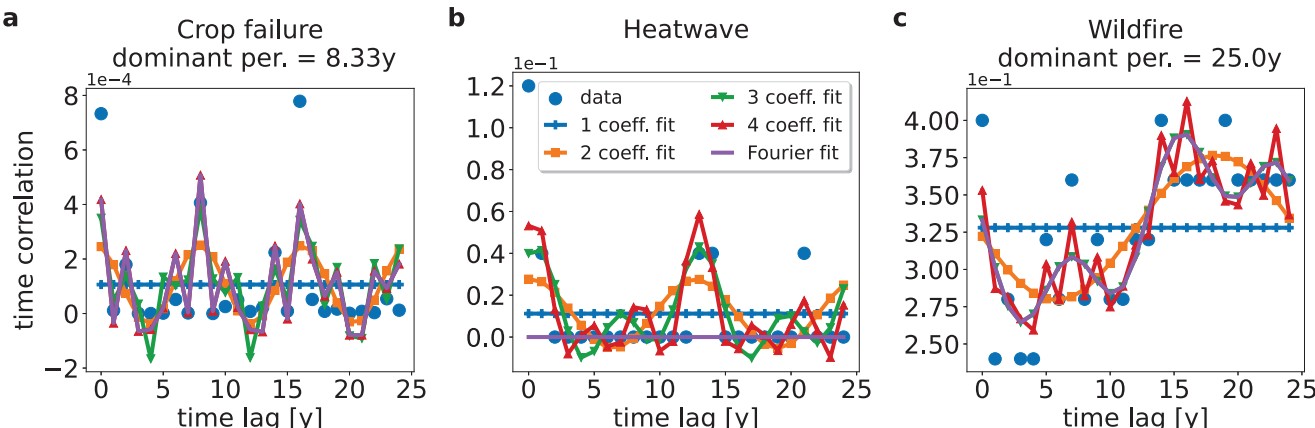

**Fig. 7 | Extreme event auto-correlation function.** Time auto-correlation function for **a** crop failure, **b** heatwave, and **c** wildfire for the time series in Fig. 6 and $t_0 = 1950$. The colored lines show Fourier fits with the 4 largest coefficients, where the violet line without marker corresponds to the Fourier fit described in step 4 of the dominant period definition. In the case where the Fourier fit fulfills the condition of step 5 and 6, we show the dominant period.

(see Supplementary Table 2). These differences highlight the difference between largest spectral peak and dominant periods.

In order to get a supplemental measure of extreme event counts within fixed time intervals we define the total affected area

$$A_{\Delta T}(t) = \sum_{\theta, \phi} \sum_{m=0}^{2\Delta T - 1} f(t + m, \theta, \phi) \cdot A(\theta, \phi), \qquad (3)$$

where $A(\theta, \phi)$ is the area of the grid cell at position $(\theta, \phi)$. Therefore, $A_{\Delta T}(t)$ measures the total area affected by an extreme event in the time window $t$ to $t + 2\Delta T$.

**Result aggregation**

Within the ISIMIP multi-model setup we aggregate the individual results for each model in terms of a median that is restricted to those model combinations where a dominant period is defined. This restriction is applied to each scenario, time window and grid cell location separately (cmp. Supplementary Discussion Section 8). This approach allows to address possible undetected dominant periods due to spectral leakage, namely changes in the Fourier spectrum depending on aspects such as the sampling of the time series. In addition, the restricted median also addresses differences between climate and impact model combinations that imply different geographical and temporal processes leading to differences in the respective extreme even time series. As a consequence we also expect differences in the geographical and temporal regularity patterns which are represented through statistical measures such as the median and standard deviation. Only in the case where no climate impact model combination detects regularity we deduce irregularity in the median result. In the case of picontrol we additionally require that a dominant period is detected in the median over all climate-impact model combinations for at least half of the time windows 1850–1899, 1900–1949, …, 2050–2099. The reasoning here is to make the picontrol runs, which are based on five time windows, comparable to the historical and SSP runs which are based on a single time window.

An estimate for the difference in dominant periods between different climate impact model combinations and time windows is given in terms of the standard deviation (see Supplementary Discussion Section 12). Note that the error estimation is limited due to the single realizations from the climate impact models. The standard deviation for all event categories is concentrated in the small range of 0–4 years.

## Data availability
The extreme event data generated in this study have been deposited in the ISIMIP database, https://doi.org/10.48364/ISIMIP.920810[76]. The input simulation data to produce the extreme events is available via https://data.isimip.org/.

## Code availability
The code used to produce the results in this manuscript is available via Code Ocean, https://doi.org/10.24433/CO.0842870.v1[77].

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

## Acknowledgements

K.Z. was financially supported through the European Union Horizon 2020 programme (HABITABLE project (Grant 869395)) and German Federal Foreign Office (PREVIEW project, Grant no. AA38220002). A.I. was partly supported by JSPS KAKENHI (Grant No. 21H05318). F.L. was supported by the National Key Research and Development Program of China (2022YFE0106500) and National Key Scientific and Technological Infrastructure project "Earth System Science Numerical Simulator Facility" (EarthLab). M.B. was financially supported through the European Union Horizon 2020 research and innovation program (FirEUrisk project, Grant no. 101003890). M.O. was supported by the Environment Research and Technology Development Fund (JPMEERF20S11805 and JPMEERF20242001) of the Environmental Restoration and Conservation Agency of Japan. S.O. was supported by the Conservation International Foundation (grant no. CI-114129). C.F. was supported by the Austrian Science Fund (FWF) (grant no. 10.55776/P36220) S.K.-G. was supported by the Natural Sciences and Engineering Research Council of Canada's Discovery Grant Program (RGPIN-2024-04188). S.R. was supported by the Helmholtz Association Impulse and Networking fund. T.I. was partly supported by the Environment Research and Technology Development Fund (JPMEERF23S21120) of the Environmental Restoration and Conservation Agency Provided by the Ministry of the Environment. W.L. was supported by the National Natural Science Foundation of China (32361143871 and 52411540183).

## Author contributions

K.Z. designed the research, developed the methodology, and performed the analysis. The interpretation of the results was done by K.Z., J.S., C.M. and S.O. with support from C.P.O.R., S.H. and J.J. K.Z. and J.S. wrote the manuscript. J.J., K.F., S.O., C.M., M.B., F.Z., S.N.G., T.I., A.I., W.L., A.K.J. and C.F. contributed and/or coordinated model experiments. C.P.O.R., J.J., K.F., S.O., C.M., M.B., F.Z., S.N.G., T.I., A.I., W.L., A.K.J., S.R., C.F., N.K., T.-S.L., S.H., and S.K.-G. contributed to the manuscript review and editing. K.Z., J.B., M.B., C.F., S.N.G., T.H., S.H., T.I., A.I., J.J., A.K.J., N.K., S.K.-G., F.L., M.L., T.-S.L., W.L., C.M., M.O., S.O., K.O., S.R., C.P.O.R., C.S., J.M.S., F.Z., K.F. and J.S. were involved in the production of the extreme event impact data.

## Funding

## Competing interests

The authors declare no competing interests.

## Additional information

[1]Potsdam Institute for Climate Impact Research, Member of the Leibniz Association, Potsdam, Germany. [2]Karlsruhe University of Applied Sciences, Karlsruhe, Germany. [3]Biodiversity and Natural Resources Program, International Institute for Applied Systems Analysis, Laxenburg, Austria. [4]School of Geography, University of Nottingham, Nottingham, UK. [5]Department of Geography, Ludwig-Maximilians-Universität, Munich, Germany. [6]Facultad de Ciencias Naturales, Universidad del Rosario, Bogotá, Colombia. [7]Institute for Agro-Environmental Sciences, National Agriculture and Food Research Organization (NARO), Tsukuba, Japan. [8]Graduate School of Agricultural and Life Sciences, The University of Tokyo, Tokyo, Japan. [9]Center for Climate Systems Research, Columbia Climate School, Columbia University, New York, NY, USA. [10]NASA Goddard Institute for Space Studies, New York, NY, USA. [11]Department of Climate, Meteorology and Atmospheric Sciences (CLiMAS), University of Illinois, Urbana-Champaign, Urbana, IL, USA. [12]Advancing Systems Analysis Program, International Institute for Applied Systems Analysis, Laxenburg, Austria. [13]School of Resource and Environmental Management, Simon Fraser University, British Columbia, Burnaby, BC, Canada. [14]International Center for Climate and Environment Sciences, Institute of Atmospheric Physics, Chinese Academy of Sciences, Beijing, China. [15]State Key Laboratory of Efficient Utilization of Agricultural Water Resources, Beijing, China. [16]National Field Scientific Observation and Research Station on Efficient Water Use of Oasis Agriculture in Wuwei of Gansu Province, Wuwei, China. [17]Center for Agricultural Water Research in China, College of Water Resources and Civil Engineering, China Agricultural University, Beijing, China. [18]NSF National Center for Atmospheric Research, Boulder, BC, USA. [19]National Institute for Environmental Studies, Tsukuba, Japan. [20]Karlsruhe Institute of Technology (KIT), Institute of Meteorology and Climate Research, Atmospheric Environmental Research (IMK-IFU), Garmisch-Partenkirchen, Germany. [21]Department of Environmental Sciences, University of Basel, Basel, Switzerland. ✉e-mail: karim.zantout@pik-potsdam.de

