## [Transparent Peer Review file · Nature Communications]

Shifting dominant periods in extreme climate impacts under global warming

Corresponding Author: Professor Karim Zantout

Version 0:

Reviewer comments:

Reviewer #1

(Remarks to the Author)

Review of "Shifting (ir)regularity regimes in extreme climate impacts under global warming" by Zantout et al.

The authors present a method to detect what they call dominant periods in time series, and apply this method to an ensemble of pi-control, present-day and future climate impact model simulations of crop failure, heatwaves and wildfires from the ISIMIP project. They find substantial changes in these periods from past to future climate and argue that their findings are relevant for adaptation planning, and more generally for predictability and identifying critical thresholds in impact systems.

While the condensed main findings are potentially interesting (although partly also trivial), I have serious concerns regarding the novelty, plausibility, soundness and relevance of the approach and findings.

Novelty:

The authors investigate dominant periods, i.e., in essence modes of variability in climate impact data, and link them qualitatively to known climatic modes of variability such as ENSO, NAO and IOD. In which way do the results add to the dozens of studies investigating the influence of modes of variability on, e.g., heatwaves or wildfires, and their changes in a warming climate? (e.g., Kenyon and Hegerl, J. Climate, 2008; Arblaster & Alexander, GRL, 2012; and all the work building upon these studies, see also the recent IPCC report, WG1, Chapter 4). I am not sure how much studies track the role of changes in modes of variability down the impact chain, but I am sure there must be plenty of research also on impacts such as crop failure.

Also, many studies so far have assessed changes in return levels of extreme events - in which way do the results better support decision making and adaptation planning than these studies? I am not convinced by the suggestions made in the introduction (see below).

I see the main novelty aspects in the global aggregate analysis in a changing climate, but the conclusions and interpretation are very weak, the whole study seems to be a more or less successful proof of concept (see below).

But most if not all results of the study could have been obtained by established methodologies in a sound way. The often pompous terminology pretends more novelty than actually can be found (in particular "temporal regularity patterns" describing nothing else than dominant frequencies).

Plausibility:

Some of the results for present (and future) climate seem to be rather implausible. For instance, it is known that wildfires in Australia and Indonesia are highly correlated to ENSO occurrence - no clear signal can be seen in the results for present climate. Also, heatwaves in Europe have caused significant impacts, such as the 2003 heatwave resulting in 10s of thousands of fatalities. In contrast, the results here suggest that heatwaves in Europe do not have any impact - not even in a

future climate. These somewhat surprising results call for a thorough evaluation against observations (and a discussion of chosen thresholds, impact definitions etc.).

Soundness:

First, I have a couple of questions regarding the developed methodology:

* How do the authors discriminate a real dominant frequency (in the sense of an absolute peak in the spectrum) from, e.g., red noise? Even without a dominant frequency, randomness may produce a non-significant peak which is misinterpreted as a dominant frequency. This holds in particular as the spectral estimate is Chi-square distributed. This is related to the choice of the largest component in the ACF - a high value does not imply quasi-periodic behaviour. Also, the chosen threshold seems to be rather arbitrary and not based on statistical reasoning.

* What do the authors mean by "aggregating over multi-model time series" (l 435)? Leakage is caused by short time series, so it seems the authors concatenate time series from different models to expand the time series. This seems rather odd to me as the approach would mix different periods from different models, thus introducing biased estimates. Usually, one would use single models and interpret model differences in the results as model uncertainty rather than mixing models. This approach may both introduce artificial periods and destroy real periods.

* Related to the question above is the following: How do the authors get robust estimates of long periods from 50-year time slices? If by concatenating different model time series, see above. If not, please explain.

* Linked to the novelty aspect above: what is the added value of this methodology to, e.g., fitting an AR2 process to the data and interpreting the main frequency of that process? The AR2 process would also provide a statistically sound framework to discriminate between quasi-periodic behaviour and red (or whatever) noise.

Second, the choice of considered impacts seems to be rather arbitrary and inconsistent. From a climate perspective, heatwaves are a meteorological phenomenon (hazard) causing impacts while wildfires and in particular crop failure are impacts beyond meteorological aspects (risks). At least, the choice should be motivated, in particular the focus on heatwaves.

Relevance:

Why is the new knowledge important? As stated above, many studies consider the role of internal variability in a changing climate, although some aspects might not have been covered yet.

In particular, I am wondering about two points raised by the authors in the introduction to motivate the relevance of their method.

First, they state that "regularity ultimately relates to the predictability of extreme climate impacts. This knowledge is relevant for insurances, disaster preparedness and response planning agencies etc." But how does the analysis here add to our understanding of predictability? As laid out above, there is plenty of knowledge about modes of variability, and their predictability has been studied in detail. Also, predictability does not only arise from quasi-periodic behaviour, but also from red noise (e.g. Hasselmann's famous Tellus paper on the stochastic climate model). Predictability in this context has been explored in much detail, and many time series models exist to predict evolutions of noisy processes.

Second, they state that "regularity of extreme climate impacts may also help to identify critical thresholds for the recovery of affected systems, e.g. when extreme climate impact frequencies are larger than typical recovery times of ecosystems". But this knowledge can be gained as well (and more appropriately) by studying return periods or waiting times - again there is plenty of statistical theory from Poisson processes to extreme value theory.

To summarise my main concerns: The manuscript contains some interesting aspects, but they have not been explored in depth as the methodology was clearly in the focus of the study. Yet the methodology is the weakest point in my opinion. So I would suggest to (1) properly cite the existing literature to be able to clearly define the starting point and added value of the study; (2) completely redo the analyses with an established methodology; and (3) discuss in depth the insights gained at the global level, ideally for a more coherent choice of impacts.

Further issues:

The title is pretentious. Please use a simple and clear language, e.g., referring to dominant periods or quasi-periodic behaviour.

In the whole introduction, I am missing a discussion of relevant literature on time series models, modes of variability, predictability and other concepts. You need to properly put your study into context, substantiate your claims and clearly explain the added value of your study/approach (see discussion above).

Line 65: Here, it is not clear what is meant by temporal irregularity patterns. Please avoid this term throughout the whole manuscript.

Line 89: Here, risks (crop failure) and hazards (heatwaves) are mixed up. Please check the IPCC risk definition (which you actually properly summarise) and apply the concept correctly. From the methods section, it seems you limit yourself to analysing the heat hazard rather than fully exploring risks arising from heatwaves (deaths, reduced workability, health impacts).

Line 93: Here, uncertainty is not limited, it is conditioned (and simply not sampled). This is a fundamental difference.

Line 146-150: The NAO is not regular and does not have preferred frequencies - why does this have a period between 2 to

15 year? Also, the named oscillations of ENSO and IOD do are interannual modes of variability. They are not the most relevant modes of variability for the listed 15 year periods. These (and longer) are typically related to decadal modes of variability such as AMO and PDO - but these are not listed here. Overall, if the authors really believe in the relevance of modes of variability, their discussion should be substantially expanded and go much deeper.

Line 97 to 116: this paragraph is very convoluted, from the discussion of regularity to the distinction between period vs. average. Please rewrite.

Line 412 to end: Please reduce the complexity of the notation. E.g., you do not need to define indices and variables for the different time periods (eq 1), and you do not need to indicate the mapping from space to space (eq 2). This terminology is confusing and might not even be known to all readers. The same holds for the terminology. E.g., the term short-time FT is used to determine changes in frequencies over time (such as in wavelet analysis). Given that you just investigate four time slices, each of them 50 years long, I would suggest to avoid the "short-time" and rather just refer to discrete FT, in particular given the audience of the journal. Also be consistent. Sometimes you use Fourier coefficient, sometimes Fourier component for your c_n (line 448 vs 449).

Author contributions: Does "supporting editing of the manuscript" constitute sufficient engagement to justify co-authorship (beyond providing data which are anyway available)?

(Remarks on code availability)

Reviewer #2

(Remarks to the Author)
See attached document

[Editorial Note: This attachment can be found at the end of this file]

(Remarks on code availability)

Version 1:

Reviewer comments:

Reviewer #1

(Remarks to the Author)

The authors have done an impressive job in addressing all my technical questions and conducted many additional analyses and partly revised their results.

The only open question now is the one on relevance: I am still not convinced about the concept of dominant periods, I did not find the arguments convincing:

(1) The argument that a classical AR framework would be computationally much more demanding (because of the model selection) is not quite correct. The classical framework intrinsically provides a theory with associated uncertainty estimates and information on the "sharpness" of the dominant frequency. Obtaining the same information from the proposed new framework would require substantial bootstrapping/Monte Carlo simulations and likely lead to the same (or even more) computational cost. This holds in particular as the model selection could be limited to 2nd order models, as only dominant periods are sought.

(2) Also the additional information compared to return periods is questionable. It would only, if it would allow for a more precise estimation of waiting times between events, but arguments for such a potential benefit are not provided by the authors. "Dominant period" does not mean that the information is strictly periodic, so no precise waiting time until an event can be derived. And given the lack of a statistical theory, prediction intervals for the waiting time cannot easily be calculated. Again, Monte Carlo simulations would be required where a classical AR2 framework might give "cheaper" information.

As the focus of the paper is still on the new concept, I see it more in a technical journal, where new analysis methods can be presented.

(Remarks on code availability)

Reviewer #2

(Remarks to the Author)
None

(Remarks on code availability)

None

REVIEWER COMMENTS

Reviewer #1 (Remarks to the Author):

Reviewer #1

Review of "Shifting (ir)regularity regimes in extreme climate impacts under global warming" by Zantout et al.

The authors present a method to detect what they call dominant periods in time series, and apply this method to an ensemble of pi-control, present-day and future climate impact model simulations of crop failure, heatwaves and wildfires from the ISIMIP project. They find substantial changes in these periods from past to future climate and argue that their findings are relevant for adaptation planning, and more generally for predictability and identifying critical thresholds in impact systems.

While the condensed main findings are potentially interesting (although partly also trivial), I have serious concerns regarding the novelty, plausibility, soundness and relevance of the approach and findings.

Our response

We thank Reviewer #1 for the detailed assessment of our manuscript.

Reviewer #1

Novelty:

The authors investigate dominant periods, i.e., in essence modes of variability in climate impact data, and link them qualitatively to known climatic modes of variability such as ENSO, NAO and IOD. In which way do the results add to the dozens of studies investigating the influence of modes of variability on, e.g., heatwaves or wildfires, and their changes in a warming climate? (e.g., Kenyon and Hegerl, J. Climate, 2008; Arblaster & Alexander, GRL, 2012; and all the work building upon these studies, see also the recent IPCC report, WG1, Chapter 4). I am not sure how much studies track the role of changes in modes of variability down the impact chain, but I am sure there must be plenty of research also on impacts such as crop failure.

Our response

We agree with Reviewer #1 that many studies have investigated the influences of climatic modes on extreme temperatures (Kenyon and Hegerl, J. Climate, 2008; Arblaster & Alexander, GRL, 2012), e.g. through statistical tests. Moreover, the IPCC report AR6 WGI Chapter 4 underlines how modes of variability (4.3.3 and 4.5.3) and temperature (4.3.4 and 4.5.1) change under SSP scenarios. In the case of climate impacts such as *historical* crop failure the WG II provides an overview (e.g. Box 5.1) on increased impact probabilities and couplings between crop yields and climate modes.

The central point of this study is not to find a connection between climate modes and extreme events as these have been established by many other studies but to investigate whether dominant periods exist in extreme climate events, quantify existing dominant periods, and showing how these change under global warming. Dominant periods are not defined in terms of correlation/synchronization to climate modes but are direct properties (Fourier component) of the extreme event time series, where a dominant period exists if at least 50% of the time series variability can be explained in terms of this dominant period (R^2 criterion). Consequently, the dominant periods result from all determinants of the extreme event time series, e.g. the interference of different climate modes but also land use patterns and crop type characteristics in the case of crop failure.

In the specific case of crop failure there are no studies on changes of dominant periods under global warming. Only one recent preprint (Harrington-Tsunogai *et al.* 2025, <https://doi.org/10.51094/jxiv.1159>) investigates how the coupling between ENSO and crop yields evolves for future scenarios (see also Introduction therein). In contrast, we focus on crop failure and investigate all sources of regularity in the time series.

We clarified the novelty of our study by expanding the literature review and contrasting our quantification of dominant periods in a harmonized multi-model setup that also includes future scenarios to previous studies.

Changes in the manuscript:

In Introduction:

“For crops, only effects from large-scale climate oscillations on crop yield variability have been studied [Iizumi2014, Ray2015, Heino2018, Anderson2019, Heino2020, IPCC AR6 WGII Chap5], such as ENSO, the Indian Ocean Dipole (IOD), and the North Atlantic Oscillation (NAO).

However, it is unclear whether such regularities of crop yield also translate into similar patterns of crop failures.

These studies on crop impacts and wildfire are based on historical observations and it remains unclear how these temporal features change under global warming.

Only one recent study investigates the relationship between ENSO and crop impacts for future scenarios [Harrington-Tsunogai2025].

In the case of heatwaves climatic oscillations have been shown to be strongly linked to the emergence of extreme heat [Kenyon2008, Arblaster2012, IPCC AR6 WGI Chap4, Domeisen2023].

Consequently, there is a strong relationship between climate oscillations and heatwave variability.

In this study, however, we investigate whether extreme event variability can be described by a single dominant period and how this period evolves under climate change.

In principle, the existence and value of the dominant period depends on the interplay of different climate modes and their interference with impact-type specific characteristics of the extreme event, e.g. land-use pattern, crop type and vegetation dynamics.” (underlined sections are new)

Reviewer #1

Also, many studies so far have assessed changes in return levels of extreme events - in which

way do the results better support decision making and adaptation planning than these studies?
I am not convinced by the suggestions made in the introduction (see below).

Our response

We thank the referee for pointing out this important difference.

The estimation of return levels is typically either based on the empirical average return levels, i.e. number of events exceeding a threshold in a given time period (e.g. Ben-Ari *et al.* Nat. Comm. 2018), or parameter estimations from (generalized extreme value) probability distributions (e.g. Evin *et al.* NHESS 2018). As such, they yield statistical estimates for return levels of extreme event but do not explicitly take the time variability of these events into account.

Our concept of dominant period is not built to replace these statistical measures but to provide an additional metric to assess whether a statistical return period of 5 years represents roughly equally distributed extreme events (regular with dominant period ~ 5 years) or non-regular random distribution (no dominant period).

A closer example for the usefulness of this concept are wildfires where the time period between severe fires is a determinant of tree regeneration properties (Fairman et al. International Journal of Wildland Fire 2016, Turner et al. PNAS 2019). Consequently, knowing dominant periods in wildfires allows to estimate regeneration risks of forests under global warming.

We added this explicit example of forest regeneration to support the usefulness of dominant periods.

Changes in the manuscript:

In Introduction:

“The first case exhibits perfect periodicity which allows for more precise disaster management while the irregularity of the second case signifies another challenge in terms of risk expectation. For example, in the case of wildfires, tree regeneration depends on the time period between severe fires and exhibits critical thresholds [Fairman2016, Turner2019].

Consequently, the recovery risks for wildfires are smaller if wildfires exhibit regularity with a dominant period above the critical threshold.

Note that dominant periods can be used in addition to average recurrence times to supply additional information on the regularity of time series.” (underlined sections are new)

Reviewer #1

I see the main novelty aspects in the global aggregate analysis in a changing climate, but the conclusions and interpretation are very weak, the whole study seems to be a more or less successful proof of concept (see below).

But most if not all results of the study could have been obtained by established methodologies in a sound way. The often pompous terminology pretends more novelty than actually can be found (in particular "temporal regularity patterns" describing nothing else than dominant frequencies).

We hope that the above extensions could clarify the aspect of novelty in the introduction of a dominant period for extreme events. Regarding the methodology, conclusion and interpretation we provide detailed answers and improvements below.

The terminology of temporal regularity was chosen since the existence of a dominant period implies that the time series variation can be well represented (R^2 criterium) by a single Fourier frequency (=regularity/periodicity) and its higher harmonic frequencies at lower amplitude. We thank Reviewer #1 for pointing out the potentially misleading terminology and agree that temporal regularity might be overinterpreted, e.g. a dominant period is not the same as a pure sin-wave without noise. Therefore, we supplemented the statistical context of this definition and clarified within the manuscript in how far the existence of a dominant period translates into statistical temporal regularity.

Changes in the manuscript:

In Abstract:

“Here, we show that under pre-industrial conditions dominant periods emerge in 28% of cropland exposed to crop failure and 10% in wildfire-affected areas, likely related to climatic oscillations such as the El Niño-Southern Oscillation, while heatwaves occur irregularly. The number of dominant periods increase by 2-13% during the transition from the pre-industrial era to the anthropocene.

In the anthropocene, the occurrence of extreme events shifts towards monotonic growth, replacing previous natural regularity patterns. Linearly de-trended projections reveal an additional shift towards smaller dominant periods due to climate change.”

In Introduction:

“While there is large agreement on the increase in intensity and frequency of extreme climate conditions, only very few studies have investigated regularity of extreme event occurrence [Castellarin2001, Yiou2004, Ghil2011, Pradhan2011, Sauer2021]. Yet, a better understanding of the temporal dynamics of extreme climate impacts is crucial for developing adaptation plans as regularity ultimately relates to the predictability of extreme climate impacts. A large class of time series approaches has been deployed to determine the characteristics of extreme events [Ghil2011], including extreme value theory [Katz2002, Haan2010] to determine return periods, spectral analysis [Yiou1996, Ghil2002] to extract periodic behavior, and stochastic or machine learning models [Schoenberg2013, Camps-Valls2025] to understand underlying processes and make predictions.

Within noisy extreme event time series, we define temporal regularity as variability that is essentially described by a single dominant period.

To this end, classic Fourier analysis in combination with statistical tools allows us to determine whether a dominant period exists in noisy extreme event time series (see details in Sec. 4).” (underlined sections are new)

“The *dominant period* is defined to be equal to the strongest periodic signal in the time series accounting for noise and its relation to other periodic signals in the time series (see Sec. 4 for more details).

Consequently, the dominant period is only defined if the time series is sufficiently well described by this single periodic signal.

This novel approach combines Fourier analysis with statistical tools and is motivated by the large size of the spatio-temporal data set within our multi-model setup, which calls for a simple characterization of regularity. (underlined words are new)

Reviewer #1

Plausibility:

Some of the results for present (and future) climate seem to be rather implausible. For instance, it is known that wildfires in Australia and Indonesia are highly correlated to ENSO occurrence - no clear signal can be seen in the results for present climate. Also, heatwaves in Europe have caused significant impacts, such as the 2003 heatwave resulting in 10s of thousands of fatalities. In contrast, the results here suggest that heatwaves in Europe do not have any impact - not even in a future climate. These somewhat surprising results call for a thorough evaluation against observations (and a discussion of chosen thresholds, impact definitions etc.).

Our response

We thank Reviewer #1 for pointing out important aspects in our definition of extreme events. We agree with Reviewer #1 that wildfires are also influenced by ENSO. In our methodology we concentrate on the dominant period which extracts the strongest mode in the time series. So far, by construction of our wildfire definition, namely aggregating to yearly burnt area with a threshold of 1% of the grid cell, we conditioned on the vegetation dynamics since the threshold is an absolute reference area and is not based on statistics (e.g. xth percentile of burnt area).

To shift the focus to climate mode contributions and harmonize the extreme wildfire definition with the other event categories we changed the definition of wildfires by using the 97.5th percentile from the pre-industrial annual burnt area distribution as threshold value for wildfire exposure.

By changing the wildfire definition, we indeed find ENSO signals in the dominant period (see Fig. 1 (c)) in accordance with literature (e.g. Chen et al. 2017). On the other hand, we lose the subdominant modes that were previously observed and related to the biome regrowth dynamics.

Similarly, the heatwaves definition that we employed was constructed for extreme *humid* heatwaves as implemented through the additional condition that extreme heat days (defined through the HWMId) also need to show humidex values larger than 45. For Europe this humidity constraint is never fulfilled.

To improve on our definition for heatwaves and harmonize the extreme event definitions we dropped this additional criterion for humidity (humidex threshold) and define extreme heatwave through the 97.5th percentile of the pre-industrial HWMId distribution.

By construction this allows for heatwaves to occur in all world regions. We checked that observed heatwave distributions are indeed in agreement with our heatwave definition. For example, the observed count (ISIMIP3a) of extreme heat impacts in 1950-1999 as derived from the new heatwave definition (see Figure below) agrees with other observations and heatwave definitions (e.g. Perkins-Kirkpatrick and Lewis 2020).

On the other hand, removing the humidity condition leads to much stronger impacts at low latitudes in the 2050-2099 time interval (see Fig. 4 (b)).

Changes in the manuscript:

(crossed out sections are removed)

In Results:

~~“Heatwaves are directly calculated from the climate model outputs based on the 97.5th percentile threshold for the daily Heat Wave Magnitude Index (HWMId) and the complementary condition that the respective heatwave days must exceed a humidity index (Humidex) value of 45.”~~

~~“In addition to the immediate temperature and humidity fluctuations the extreme heatwave definition that involves conditions on both HWMId and Humidex further reduces the probability of a heatwave under pre-industrial climate conditions.”~~

~~“Exposure to heatwaves and therefore dominant periods only appear at low and mid latitudes where we also encounter high humidity with typical values for dominant periods between 7 and 13 years appearing in close to 1% of all affected areas.”~~

We moved the wildfire section under pre-industrial conditions (previous definition) to Supplemental Material Section 5 and replaced the section by the following: “Wildfire dynamics involve several time scales related to climate oscillations and different vegetation growth dynamics [Archibald2013, Chen2017].

We define extreme wildfire through a threshold on the annual burnt area given by the 97.5th percentile of the pre-industrial distribution (see Sec. Methods).

We observe more irregularity in wildfires than regularity with 90% of all impacted grid cells exhibiting no dominant periods.

We observe dominant periods in the range of 4 to 13 years, similar to the ones for crop failures and heatwaves. These occurrences of dominant periods appear in all world regions in agreement with previous studies showing an influence of climate modes on wildfires [Chen2017, Shi2022].

The largest cluster is observed in South America where previous studies detected that El Niño2015/16 led to the largest fire response [Burton2020].

The correlation coefficient between global extreme wildfire affected area and the Southern Oscillation Index is 0.22 (see Supplementary Fig. 8 c) which is smaller than the respective value for heatwaves and consistent with wildfire dynamics being dependent not only on climatic factors but also on the biome types [Archibald2013]. Our definition of extreme wildfire does not resolve the vegetation regrowth dynamics.

To extract the influence of the different biome types we applied a different classification of wildfires based on area thresholds in Supplementary Material Sec. 5 and dominant periods that are consistent with observed wildfire dynamics.”

Similarly, we changed the heatwaves section to:

“Here again the regime shift around 1950 is obvious in the total affected area (Fig. 4 b) which leads to the emergence of regularity in the transitioning phase.

Regularity occurs since the probability for heatwaves increases with global warming while the heatwave expectation value under picontrol was one extreme event in 40 years.

Increases in temperature variability due to human forcing were already observed in similar modeling setups [Olonscheck2021] and support our interpretation of the results in the transition phase.

The emergent regularity patterns are similarly distributed to the crop failure dominant periods with highest occurrences of dominant periods in the range of 10 to 13 years. and we find heatwave clusters in all world regions.

This picture changes again drastically in the 2040-2069 period where dominant periods can be determined in almost all regions with a clear signal from dominant periods of 1-4, 13-16 years or even more years (see Supplementary Fig. 30 a).

The regularities from 1-4 years are saturated values where the picontrol threshold values is exceeded almost every year (see Supplementary Fig. 23 b) while largest dominant periods speak of a clear warming trend that dominate the time series.

Highest frequencies at low latitudes are consistent with previous studies of future scenarios on heat indicators [Fan2020, Schwingshackl2021].

After linear detrending (see Fig. 4 (c)) we find a resurgence of irregularity (40% in the total distribution) and prevailing prevailing highest frequency clusters between 1 and 4 years (red) across all world regions.

This increase towards smallest dominant periods may be not only related to the general increase in temperature within SSP5-8.5 but also to stronger ENSO effects with a fingerprint in the 2 to 7 years range.”

~~“Moreover, we observe an extension of affected areas compared to the pre-industrial climate scenario due to anthropogenic forcing (compare Fig. 4 (b)).”~~

In Methods:

~~“Finally, we calculate heatwaves from near-surface air temperature and relative humidity based on the Heat Wave Magnitude Index daily (HWMId) [Russo2015, Russo2017] and on the Humidex [Masterton1979].~~

~~Specifically, we define a grid cell to be exposed to a heatwave if the HWMId of that year exceeds the 97.5th percentile of the picontrol distribution of that grid cell. and if the Humidex exceeds a value of 45 on all hot days corresponding to the HWMId threshold of that year.~~

~~While heatwaves can occur below a Humidex value of 45 in some world regions our global analysis necessitates a globally applicable threshold that is severe in all world regions.”~~

Reviewer #1

Soundness:

First, I have a couple of questions regarding the developed methodology:

* How do the authors discriminate a real dominant frequency (in the sense of an absolute peak in the spectrum) from, e.g., red noise? Even without a dominant frequency, randomness may produce a non-significant peak which is misinterpreted as a dominant frequency. This holds in particular as the spectral estimate is Chi-square distributed. This is related to the choice of the largest component in the ACF - a high value does not imply quasi-periodic behaviour. Also, the chosen threshold seems to be rather arbitrary and not based on statistical reasoning.

Our response

We thank Reviewer #1 for raising this important aspect. We agree that peaks in the Fourier spectrum may not be actual frequencies of the underlying process but statistical artefact from typical red or white noise (Hasselmann 1976, Mann and Lees 1996). This is also true for our methodology that is based on (short-time) Fourier analysis.

So far, we have performed a single test for white noise on the overall data in Supplemental Material Sec. 3 but indeed short-range correlation can induce peaks at low frequencies that do not correspond to true peaks (red noise).

To test our results against red noise we extend our methodology with an additional red noise test based on an AR(1) process and the respective theoretical power spectrum (Gilman et al. 1963, Torrence and Compo 1998) with a significance level of 5%.

The number of significant signals indeed decreased by around 30% in the case of crop failure while the distribution of dominant periods did not change significantly.

We chose the R^2 criterium based on the statistical interpretation that this value represents, namely the fraction of variation that is explained by a model. In our case we calculate the fraction of variation that is explained by the dominant period and its higher periods. Therefore, the criterium of $R^2 > 0.5$ implies that the majority of the variation is explained by the dominant period and its harmonics. To prove the robustness of this threshold we added the R^2 test in the Supplementary Discussion Sec. 1.2.

We clarified the motivation for the R^2 criterion in the manuscript.

Changes in the manuscript:

In Methods:

“6. For the dominant period candidate $\text{per}(c'_0)$ we test if the respective peak $|c'_0|^2$ in the power spectrum is due to red noise [Torrence1998, Zhang2011] with a significance level of 5%. The red noise spectrum $|c_n|^2$ within an AR(1) model is distributed as

$$|c_n|^2 \sim \frac{1 - \varphi^2}{1 + \varphi^2 - 2\varphi \cos \frac{2\pi n}{\Delta T}} \chi_2^2,$$

where φ is the lag-1 correlation function and χ_2^2 is the chi-squared distribution with 2 degrees of freedom.”

~~“To partially absorb the effects of stochasticity in the definition of the dominant period we introduce an adjusted R^2 criterion in step 5 and make no restrictions on possible phases between the different Fourier components.~~

~~This means that missing periodic signals in some periods are tolerated to an extent determined by the adjusted R^2 criterion.~~

The R^2 criterion in step 5 accounts for the amount of variability explained by the dominant period. If the criterion is fulfilled we can state that at least half of the variability is explained by the dominant period and its higher harmonics.

In this sense the criterion makes sure that the dominant period is indeed significant and additionally account for stochasticity of the time series.” (crossed out section is replaced)

“As geophysical time series typically contain red and white noise [Hasselmann1976, Mann1996] we perform a significance test in step 6 to make sure that the detected dominant periods are not noise artifacts [Torrence1998, Zhang2011].”

Reviewer #1

* What do the authors mean by "aggregating over multi-model time series" (l 435)? Leakage is caused by short time series, so it seems the authors concatenate time series from different models to expand the time series. This seems rather odd to me as the approach would mix different periods from different models, thus introducing biased estimates. Usually, one would use single models and interpret model differences in the results as model uncertainty rather than mixing models. This approach may both introduce artificial periods and destroy real periods.

Our response

We thank Reviewer #1 for pointing out the imprecise formulation. We do not concatenate time series from different models but indeed calculate dominant periods for all models separately and then calculate the median and standard deviation of those dominant periods.

Since spectral leakage may occur due to the short time series we use the median and standard deviation to get more robust estimates of the true dominant period.

We clarified the formulation accordingly.

Changes in the manuscript:

In Methods:

~~“Additionally, we aggregate over multi-model time series results which partly accounts for known caveats such as spectral leakage.~~

Additionally, we use the median and standard deviation from the multi-model distribution of dominant features to partly account for known caveats such as spectral leakage (see Sec. 4.3).”

Reviewer #1

* Related to the question above is the following: How do the authors get robust estimates of long periods from 50-year time slices? If by concatenating different model time series, see above. If not, please explain.

Our response

The total length of our time series are 251 years (1850-2100) for each model and scenario combination. As such, we are limited in resolving largest dominant periods by the length of the time series. In the case of the picontrol scenario we make use of stable climatic conditions and calculate dominant periods on the full time series for all models separately. We then calculate the multi-model median to estimate robust dominant periods (Supplementary Material Sec. 4).

In the case of SSP scenarios we have no stable climatic conditions and are restricted to the 50-year time windows. Here, we similarly perform the multi-model median for each time window separately but do not concatenate any time series (see response above).

Reviewer #1

* Linked to the novelty aspect above: what is the added value of this methodology to, e.g., fitting an AR2 process to the data and interpreting the main frequency of that process? The AR2 process would also provide a statistically sound framework to discriminate between quasi-periodic behaviour and red (or whatever) noise.

Our response

Our methodology is based on a short-time Fourier analysis with an additional new test for red noise (see response above). The multi-model setup with $0.5^\circ \times 0.5^\circ$ spatial resolution demands a fast computation of the power spectrum with little or no hyper-parameters that need to be adjusted. As such, Fourier analysis with its Fast Fourier Transform implementation allows for an established and fast implementation. By analysing the resulting Fourier coefficients and performing statistical tests - R^2 condition to determine a *dominant* period and new red noise to check significance (see above)- we are able to find robust dominant periods.

On the other hand, there are several time series approaches that are more elaborate but typically involve hyperparameters that need to be estimated for each impact, scenario, model, or event region/location.

In addition, the resulting power spectrum of such approaches is not discrete but continuous which makes it difficult to determine a single dominant period based on peaks and their relationship in terms of harmonics (see examples below).

In the specific case of AR(n) models the hyperparameter is the set of lags to include. AR(2) implies that only lag-1 and lag-2 contributions enter the auto-regressive equation but we cannot exclude longer correlation effects, e.g. biome regrowth dynamics. The optimal set of lags can be determined through the AIC information criterion by comparing all possible model setups (as implemented in the Python library statsmodels.tsa.ar_model.ar_select_order).

Since this approach is computationally very demanding we compare the dominant period estimates from the Fourier analysis to the AR(n) estimates (with optimal set of lags) for the exemplary locations from Fig. 7:

	Dominant period (Fourier analysis)	Strongest period (Optimal AR(n) model)
Example 1 (Fig. 7 (a))	8.33y	15.53y
Example 2 (Fig. 7 (b))	None (12.5y candidate)	6.88y
Example 3 (Fig. 7 (c))	25y	7.63y

In the first example our method (step 4) recognizes the smaller peak at time lag 8 and the relationship to the largest peak at time lag 16 as a higher harmonic period (see Fig. 7 (a)). Consequently, the dominant period is given by 8 years, while the AR(n) model correctly identifies the largest peak at time lag 16 but there is no trivial way to connect this peak to the other spectral peaks (harmonics) of the spectrum.

The second example shows that both methods find different dominant periods in an ambiguous case, where peaks appear at 0, 13,14, and 21 years lag (see Fig. 7 (b)). These peaks are compatible with dominant periods of 12.5 and 6.88 years where in both cases one peak is misplaced (21 instead of 25) or missing (6.88 missing). In the case of 12.5 years the significance test for red noise fails in this ambiguous case which is why our method finds no dominant period.

In the last example we find different periods due to differences in the spectral estimates. The largest spectral density in Fourier analysis is found to be at 25y period in contrast to 7.63 years from the AR(n) model. This difference stems from the trend in the time series which is partly absorbed in the coefficients of the AR model while the Fourier analysis absorbs the trend mostly in the largest dominant period.

Changes in the manuscript:

In Methods:

~~“Note that more advanced approaches [Ghil2002, Ghil2011] such as the Multi Taper method [Thomson1982, Privalsky2018] allow to minimize spectral leakage and reduce variance at optimized spectral resolution, but they yield a complex power spectrum that depends on the specific configuration of the model.~~

~~For our purposes we can rely on a simpler quantification of regularity as we are only interested in the dominant spectral feature.~~

Note that more advanced approaches [Thomson1982, Ghil2002, Privalsky2018, Ghil2011] such as the Multi Taper method allows to minimize spectral leakage and reduce variance at optimized spectral resolution while autoregressive models can explicitly account for red noise. On the other hand, these elaborate approaches typically involve higher computational costs and hyperparameters that need to be estimated for each impact, scenario, model, or even region/location.

In addition, the resulting power spectrum of such approaches is not discrete but continuous which makes it difficult to determine a single dominant period based on peaks and their relationship in terms of harmonics.”

“A comparison of dominant periods in Fig. 7 and dominant periods estimated from the largest spectral peak of autoregressive models shows differences due to the additional steps 4-6 in our methodology and differences in the spectral estimates themselves (see Supplementary Table 2).

These differences highlight the difference between largest spectral peak and dominant periods.”

New Supplementary Material Sec. 9:

“Comparison to autoregressive model

Here, we show the difference between the dominant period as calculated from Fourier analysis described in Sec. 4 with results from autoregressive models (AR).

Since autoregressive models produce a continuous power spectrum we will compare to the strongest peak which corresponds to the strongest period.

In addition, we estimate the optimal set of lag parameters for the autoregressive model through the AIC information criterion by comparing all possible model setups as implemented in `ar_select_order` method of the AR class in statsmodels [statsmodels].

	Dominant period (Fourier analysis)	Strongest period (Optimal AR(n) model)
Example 1 (Fig. 7 a)	8.33y	15.53y
Example 2 (Fig. 7 b)	None (12.5y candidate)	6.88y
Example 3 (Fig. 7 c)	25y	7.63y

Table 2: Comparison between dominant period and strongest period.

In the first example our method (see Sec. 4 step 4) recognizes the smaller peak at time lag 8 and the relationship to the largest peak at time lag 16 as a higher harmonic period (see Table 2). Consequently, the dominant period is given by 8 years, while the AR(n) model correctly identifies the largest peak at time lag 16 but there is no trivial way to connect this peak to the other spectral peaks (e.g. harmonics) of the spectrum.

The second example shows that both methods find different dominant periods in an ambiguous case, where peaks appear at 0, 13,14, and 21 years lag.

These peaks are compatible with dominant periods of 12.5 and 6.88 years where in both cases one peak is misplaced (21 instead of 25) or missing (6.88 missing).

In the case of 12.5 years the significance test for red noise fails in this ambiguous case which is why our method finds no dominant period.

In the last example we find different periods due to differences in the spectral estimates.

The largest spectral density in Fourier analysis is found to be at 25y period in contrast to 7.63 years from the AR model.

This difference stems from the trend in the time series which is partly absorbed across coefficients of the AR model while the Fourier analysis absorbs the trend mostly in the largest dominant period.”

Reviewer #1

Second, the choice of considered impacts seems to be rather arbitrary and inconsistent. From a climate perspective, heatwaves are a meteorological phenomenon (hazard) causing impacts while wildfires and in particular crop failure are impacts beyond meteorological aspects (risks). At least, the choice should be motivated, in particular the focus on heatwaves.

Our response

We have adjusted the definition of extreme events (see above) to make the choices consistent with each other and based on percentile thresholds from the picontrol reference. We agree with Reviewer #1 that crop failure and wildfire are determined by both meteorological and other factors. As such, we included these impacts to investigate changing patterns in dominant periods that go beyond meteorological aspects while fixing socio-economic conditions to ensure consistency. Remaining on the hazard side and investigating only changing temperature patterns miss important dimensions on the risk side such as buffer systems and intrinsic time scales (e.g. regrowth dynamics in wildfires, crop type mix at each grid cell). We clarified this motivation in the manuscript.

Changes in the document:

In Introduction:

“Here, we consider three types of extreme climate ~~impacts~~ events, namely crop failures, heatwaves, and wildfires.

While the definition of heatwaves only depends on temperature, crop failures and wildfires are derived from impact models that combine the temporal evolution of climatic variables with soil and vegetation characteristics. Consequently, heatwaves correspond to a meteorological hazard whereas crop failures and wildfires are impacts. The inclusion of both allows us to investigate direct climate change effects (heatwaves) and the interplay with impact related dynamics (crop failures and wildfires).” (underlined sections are new)

Reviewer #1

Relevance:

Why is the new knowledge important? As stated above, many studies consider the role of internal variability in a changing climate, although some aspects might not have been covered yet.

In particular, I am wondering about two points raised by the authors in the introduction to motivate the relevance of their method.

First, they state that "regularity ultimately relates to the predictability of extreme climate impacts. This knowledge is relevant for insurances, disaster preparedness and response planning agencies etc." But how does the analysis here add to our understanding of predictability? As laid out above, there is plenty of knowledge about modes of variability, and their predictability has been studied in detail. Also, predictability does not only arise from quasi-periodic behaviour, but also from red noise (e.g. Hasselmann's famous Tellus paper on the stochastic climate model). Predictability in this context has been explored in much detail, and many time series models exist to predict evolutions of noisy processes.

Our response

We agree with Reviewer #1 that there is a vast field of research predicting and assessing the quality of climate time series. Especially in the case of stochastic climate models there are various ways to study uncertainties of model predictions.

In our study we focus on climate impacts that are derived from deterministic process-based impact models within the unified modelling protocol ISIMIP that receive climate input from GCMs. The only exception are heatwaves which are directly derived from GCM output. To the knowledge of the authors ISIMIP is the only modelling framework that produces harmonized climate impact data across impact categories which is essential for the intercomparability. Moreover, there are no studies on crop failure and wildfire that study how dominant periods or regularities change under global warming (see also response above).

Due to the limitation in the number of models, time resolution (yearly data from 1850 to 2100) and the deterministic character of impact models it is not straight-forward to assess how climate variability translates to climate impact variability and the prediction intervals can be very broad (see S. Lange *et al.* Earth's Future 2020).

Within this study we analyse changing patterns of dominant periods within harmonized impact data which may be attributed to different stochastic and deterministic processes that arise from the GCMs and the impact models, respectively, while impact studies typically consider average

return periods from statistical approaches (see response above). Our study supplies an additional estimate whether these return periods appear regularly or not. We thank Reviewer #1 for point out potentially misinterpreted signals from red noise and adjusted the methodology accordingly (see response above).

In addition to the previous manuscript changes we clarified the relevance of our work by mentioning the additional time variance effects from the impact models.

Changes in the manuscript:

In Introduction:

~~“Here, we present a novel method to analyze regularities of extreme climate impacts using~~ Our study is based on the latest climate model projections to drive an ensemble of climate impact models from the Inter-Sectoral Impact Model Intercomparison Project (ISIMIP). ISIMIP assembles harmonized inputs and provides consistent modeling protocols for a multi-model climate impact framework [Frieler2024]. We use global gridded time series data from ISIMIP phase 3b ~~in combination with extreme event definitions from Ref. [Lange2020]~~ to calculate crop failure, heatwave and wildfire time series from 1850 to 2100. The extreme events are defined ~~either~~ through percentiles of the respective pre-industrial extreme event indicator distribution ~~or direct outputs from the impact models~~ (see Sec. 4 for details). The combination of climate and impact models allows us to investigate changes in the time series variability not only due to changes in climate indicators but also their effects within impact models.” (underlined sections are new, crossed out sections are removed)

Reviewer #1

Second, the state that "regularity of extreme climate impacts may also help to identify critical thresholds for the recovery of affected systems, e.g. when extreme climate impact frequencies are larger than typical recovery times of ecosystems". But this knowledge can be gained as well (and more appropriately) by studying return periods or waiting times - again there is plenty of statistical theory from Poisson processes to extreme value theory.

Our response

We agree with Reviewer #1 that statistical tools to estimate return periods of extreme events are fundamentally important for investigating recovery risks for affected systems. Our notion of dominant periods is not a replacement but a supplement to these fundamental statistics, measuring whether statistical return periods also appear as dominant frequencies in the time series. We extended our motivation by another example on wildfires (see response above).

Statistical approaches typically involve assumptions on the underlying process that are critical for the return period estimate [Ghil2011, Sillmann2017]. For example, Poisson (point) processes either assume independent constant rates/intensities (homogenous) or rate functions (heterogenous) that depend on time, location, model etc. The estimation of this parameter is difficult due to the single realization for each time-location-model-impact-scenario combination in ISIMIP (except for the picontrol setup where we have static conditions). Within a stochastic setup we would be able to fit several realizations to a Poisson process and extract the underlying parameter but since impact models are computationally very demanding they produce single realizations.

In addition, estimates of return periods from probability distributions face further challenges from restricted time length in the case of wildfires [Archibald et al. PNAS 2013] or bad fits in the case of GEV distribution in the case of floods [Zhou et al. NHESS 2021].

We extend our Methods section by a literature discussion on statistical approaches.

Changes in the manuscript:

In Methods:

“Furthermore, statistical approaches in time series analysis [Ghil2011, Sillmann2017] typically involve assumptions on the underlying process that are critical for the return period estimate. For example, Poisson processes are based on (heterogenous) rates that may depend on time, location, model etc. Within the ISIMIP setup this parameter is difficult to estimate as impact time series are provided as single realizations. Therefore, additional assumptions or approximations are needed to get enough data for fitting to a Poisson process. Estimates of return periods from probability distributions face these challenges from restricted time length in the case of wildfires [Archibald2013] and in the case of generalized extreme value distributions of floods [Salas2014, Zhou2021].”

Reviewer #1

To summarise my main concerns: The manuscript contains some interesting aspects, but they have not been explored in depth as the methodology was clearly in the focus of the study. Yet the methodology is the weakest point in my opinion. So I would suggest to (1) properly cite the existing literature to be able to clearly define the starting point and added value of the study; (2) completely redo the analyses with an established methodology; and (3) discuss in depth the insights gained at the global level, ideally for a more coherent choice of impacts.

Our response

We thank Reviewer #1 for the valuable suggestions. We believe that our revised manuscript, which includes an expanded literature review, an extended methodology, the incorporation of AR(n) model comparisons, improvements in the definitions of extreme events, and a more comprehensive discussion of the results (detailed in our responses below), effectively addresses the main concerns.

Reviewer #1

Further issues:

The title is pretentious. Please use a simple and clear language, e.g., referring to dominant periods or quasi-periodic behaviour.

Our response

We thank Reviewer #1 for the suggestion and changed the title to “Shifting dominant periods in extreme climate impacts under global warming”.

Reviewer #1

In the whole introduction, I am missing a discussion of relevant literature on time series models, modes of variability, predictability and other concepts. You need to properly put your study into context, substantiate your claims and clearly explain the added value of your study/approach (see discussion above).

Our response

We thank Reviewer #1 for the suggestion and improved (in addition to the above-mentioned changes) on the methodological literature.

Changes in the manuscript:

In Introduction:

“A large class of time series approaches has been deployed to determine the characteristics of extreme events [Ghil2011], including extreme value theory [Katz2002, Haan2010] to determine return periods, spectral analysis [Yiou1996, Ghil2002] to extract periodic behavior, and stochastic models [Schoenberg2013] to understand underlying processes and make predictions.

~~In the case of extreme climate impacts we define temporal regularity if the temporal variation is essentially described by a dominant period.~~

Within noisy extreme event time series, we define temporal regularity as variability that is essentially described by a dominant period. To this end, classic Fourier analysis in combination with statistical tools allows us to determine whether a dominant period exists in noisy data (see details in Sec. 4).” (crossed out sentence is removed)

Reviewer #1

Line 65: Here, it is not clear what is meant by temporal irregularity patterns. Please avoid this term throughout the whole manuscript.

Our response

We thank Reviewer #1 for pointing out the unclear formulation. We replaced the expression with “regularity of extreme event occurrence”.

Reviewer #1

Line 89: Here, risks (crop failure) and hazards (heatwaves) are mixed up. Please check the IPCC risk definition (which you actually properly summarise) and apply the concept correctly. From the methods section, it seems you limit yourself to analysing the heat hazard rather than fully exploring risk s arising from heatwaves (deaths, reduced workability, health impacts).

Our response

We thank Reviewer #1 for noticing this inconsistency. We clarified that terminology and discuss the choice of event types (see response above).
Indeed, in the case of heatwaves we discuss only the hazard component since the risk component, e.g. mortality, is still on-going work within ISIMIP.

Changes to the manuscript:

In Introduction:

“Consequently, heatwaves are restricted to the hazard component as the full impact modelling, e.g. heat-related mortality, within ISIMIP is ongoing work.” (underlined sentence is new)

Reviewer #1

Line 93: Here, uncertainty is not limited, it is conditioned (and simply not sampled). This is a fundamental difference.

Our response

We thank Reviewer #1 for noticing this error. We clarified the sentence accordingly.

Changes in the manuscript:

In Introduction:

~~“To limit the uncertainty from socio-economic factors and investigate regularity patterns that exclusively arise from natural processes such as vegetation growth dynamics as well as temperature and precipitation variability, we here keep direct human influences such as land use changes and land management constant at 2015 conditions.”~~

Reviewer #1

Line 146-150: The NAO is not regular and does not have preferred frequencies - why does this have a period between 2 to 15 year? Also, the named oscillations of ENSO and IOD do are interannual modes of variability. They are not the most relevant modes of variability for the listed 15 year periods. These (and longer) are typically related to decadal modes of variability such as AMO and PDO - but these are not listed here. Overall, if the authors really believe in the relevance of modes of variability, their discussion should be substantially expanded and go much deeper.

Our response

We thank Reviewer #1 for highlighting the misleading formulation. Indeed, NOA does not have a preferred frequency and its variability ranges from interannual to multidecadal scales (Wanner et al. 2001). Previous studies established links between crop yields and NOA in certain world regions (Heino et al. 2018, Anderson et al. 2019, Heino et al. 2020). Consequently, we assume that NOA may plausibly also contribute to the dominant period of crop failure. Its contribution is not directly linked to a fixed frequency but its decadal variation may modulate dominant periods in the range of 2-15 years.

While ENSO indeed shows interannual modes of variability it also exhibits spectral peaks at 2-8 years (Kestin et al. 1998). In the case of IOD we agree that the spectral modes are high-frequency but IOD is correlated with ENSO (Behera et al. 2006, Stuecker et al. 2017) and influences crop yield variability (Heino et al. 2018, Anderson et al. 2019, Heino et al. 2020). In the case of AMO and PDO that show prominent low frequency modes of variation we find no global impact on crop production but only regional effects (Schillerberg and Tian 2020, Xu et al. 2020).

We clarified the difference in the influences of the climate modes.

To further investigate the connection between ENSO and crop failure (and heatwaves and wildfire) within our modelling framework we perform a correlation analysis. We use the annual total area affected by an extreme event from observation-based ISIMIP3a simulations and the annual average Darwin Southern Oscillation Index based on annual standardization as ENSO index to account for the data quality in Tahiti before 1935 [Climate Analysis Section, NCAR, Boulder, USA, Trenberth (1984)].

As expected heatwaves show the strongest correlation (0.37) with ENSO while for wildfires we observe a slightly smaller value of 0.22. On the other hand, for crop failure we find no correlation (0.02) which is due to the strong region and crop type dependence on climate variations (Heino et al. 2018, Anderson et al. 2019). By calculating crop type specific correlations, we obtain a clearer picture with positive median correlation for maize and negative median correlation for wheat in agreement with studies on crop yield variability (Iizumi et al. 2014) while in the case of rice and soy we find a median positive and near-zero correlation, respectively, which is opposite to the crop yield variability studies (Iizumi et al. 2014). In summary, we indeed find correlations between the extreme events and ENSO which in the case of crop failure become evident when considering crop resolved correlation.

Changes in the manuscript:

In results:

“A likely explanation for the ~~regularity patterns~~ dominant periods are influences from ENSO, IOD, and North Atlantic Oscillation (NAO) [Heino2018, Anderson2019, Heino2020]. Note that NOA exhibits no clear low-frequency regularities but decadal variations [Wanner2001] that may modulate dominant periods in the observed ranges. Similarly, IOD exhibits strong high-frequency modulations that cannot explain the observed dominant periods but IOD is correlated with ENSO [Behera2006, Stuecker2017] and may therefore influence the observed dominant periods.

~~These natural climate oscillations have frequencies of about 2 to 15 years and are~~ On the other hand, ENSO shows oscillations that are in the range of 2-8 years [Kestin1998] and in addition to NOA and IOD is known to affect regional climate, and thereby also crops, across the globe [Iizumi2014, Ray2015, Heino2018, Anderson2019, Heino2020].” (underlined sections are new, crossed out sections are removed)

“Correlation analysis indeed shows near-zero correlation between aggregated crop failure affected areas and the Southern Oscillation Index while crop specific analysis reveals non-zero model-median correlation in the case of maize, rice, and wheat (see Supplementary Material Fig. 8 a and 9).”

“Moreover, other climate modes that exhibit decadal variation such as the Pacific Decadal Oscillation (PDO) and Atlantic Multidecadal Oscillation (AMO) have been shown to

regionally affect crop yields [Schillerberg2020, Xu2021] and therefore may locally influence dominant periods.”

In Supplementary Material:

“Correlation analysis

To investigate the connection between ENSO and the different extreme event categories we perform a correlation analysis.

We use the annual extreme event affected area from historical ISIMIP3a simulations (see Supplementary Material Sec. 1.3) and the yearly-mean Darwin Southern Oscillation Index (SOI) based on annual standardization [Trenberth1984].

Note that this choice is due to quality issues from Tahiti data before 1935 [Trenberth1984].

Fig. 8: **Correlation between extreme event and Southern Oscillation Index.** Pearson correlation between historical annual extreme event affected area from ISIMIP3a simulations and yearly-mean Darwin Southern Oscillation Index (SOI) for (a) crop failure, (b) heatwave, and (c) wildfire.

As expected heatwaves show the strongest correlation (0.37) with ENSO while for wildfires we observe a slightly smaller value of 0.22 (see Fig. 8 b and c).

On the other hand, for crop failure we find no correlation (0.02) which is due to the strong region and crop type dependence on climate variations [Heino2018, Anderson2019] (see Fig. 8 a).

Figure 9: **Crop resolved correlation.** Crop resolved Pearson correlation between historical annual crop failure affected area from ISIMIP3a simulations and yearly-mean Darwin Southern Oscillation Index (SOI).

By calculating crop type specific correlations, we obtain a clearer signal with positive model-median correlation for maize and negative model-median correlation for wheat in agreement with studies on crop yield variability [Iizumi2014] (see Fig. 9).

In the case of rice and soy we find a model-median positive and near-zero correlation, respectively, which is opposite to the crop yield variability study from Ref. [Iizumi2014] where crop yields are considered.”

Reviewer #1

Line 97 to 116: this paragraph is very convoluted, from the discussion of regularity to the distinction between period vs. average. Please rewrite.

Our response

We agree that this paragraph is very dense and simplified it accordingly.

Changes in the manuscript:

In Introduction:

“The *dominant period* is defined to be equal to the strongest periodic signal in the time series accounting for noise and its relation to other periodic signals in the time series (see Sec. 4 for more details). Consequently, the dominant period is only defined if the time series is sufficiently well described by this single periodic signal.

This approach is motivated by the large size of the spatio-temporal data set within our multi-model setup, which calls for a simple characterization of regularity.

For example, the detection of a dominant period of 2 years indicates the prevalence of 2 years recurrence time in extreme event exposure. Both time series (1,0,1,0,1,0,1,0) and (1,1,0,0,0,1,0,1) have average recurrence time $8/4 = 2$ since we observe 4 (1: event, 0: no event) events in 8 time steps. On the other hand, the dominant period is 2 in the first (periodic) case and undefined in the second (irregular) case. The first case exhibits perfect periodicity which allows for more precise disaster management while the irregularity of the second case signifies another challenge in terms of risk expectation.

This knowledge is relevant for insurances, disaster preparedness and response planning agencies, etc.

In addition, regularity of extreme climate events may also help to identify critical thresholds for the recovery of affected systems, e.g. when extreme climate impact frequencies are larger than typical recovery times of ecosystems [VanLangevelde2003, Osborne2017].

For example, in the case of wildfires, tree regeneration depends on the time period between severe fires and exhibits critical thresholds [Fairman2016, Turner2019].

Consequently, the recovery risks for wildfires are smaller if wildfires exhibit regularity with a dominant period below the critical threshold.

Note that dominant periods can be used in addition to average recurrence times to supply additional information on the regularity of time series.”

Reviewer #1

Line 412 to end: Please reduce the complexity of the notation. E.g., you do not need to define indices and variables for the different time periods (eq 1), and you do not need to indicate the mapping from space to space (eq 2). This terminology is confusing and might not even be known to all readers. The same holds for the terminology. E.g., the term short-time FT is used to determine changes in frequencies over time (such as in wavelet analysis). Given that you just investigate four time slices, each of them 50 years long, I would suggest to avoid the "short-time" and rather just refer to discrete FT, in particular given the audience of the journal. Also be consistent. Sometimes you use Fourier coefficient, sometimes Fourier component for your c_n (line 448 vs 449).

Our response

We thank Reviewer #1 for the suggestion and simplified the notation accordingly.

Reviewer #1

Author contributions: Does "supporting editing of the manuscript" constitute sufficient engagement to justify co-authorship (beyond providing data which are anyway available)?

Our response

We thank Reviewer #1 and corrected the author contribution.

Reviewer #2

Reviewer #2 (Remarks to the Author):

Review for Nature Communications NCOMMS-24-85559

Title: "Shifting (ir)regularity regimes in extreme climate impacts under global warming"

Authors: Karim Zantout et al.

I find the study quite interesting for publication. I hope that my requests, comments and concerns can be answered satisfactorily by the authors, and under the consideration of the editorial board of the journal.

Our response

We thank Reviewer #2 for carefully assessing our manuscript and the positive feedback.

Reviewer #2

1. General comments

1. Abstract: “The”, as a generic expression of extreme climate events seems to be confusing and potentially leading to misunderstanding when reading the abstract. I would suggest “Several”, “Plenty”, “Many”, or perhaps nothing, and so state directly “Spatio-temporal patterns”. And of course, state that the work is going to focus on three examples or specific events: crop failures, heatwaves and wildfires. Otherwise, it could be thought that it can be applied/generalized to other, among many, extreme events: droughts or dry spells, heavy rains, wind extremes, coldwaves, etc..

Our response

We thank Reviewer #2 for pointing out this misleading formulation.

Changes in the manuscript:

Abstract:

“~~The~~ Spatio-temporal patterns of extreme climate impacts have been extensively studied, yet two questions remain underexplored: Do such events occur regularly, and how do regularity patterns change under global warming? We address these questions by investigating dominant periods in crop failure, heatwaves, and wildfire data.”
(crossed-out section is removed, underlined section is new)

Reviewer #2

2. Line 65: Although it is not fully related to heatwaves temporal regularity patterns, the work of Molina et al., (2020), <https://doi.org/10.1038/s41598-020-65663-0>, figures 4-5 show time trends over several subregions over Europe, and so it could be named on that list of references, and also on the comment at line 81. When mentioning heatwaves characterization, I strongly suggest also some of the many works led by Sarah Perkins, known as one of the main researchers on these extreme events. See for example, on this same journal, Perkins-Kirkpatrick and Lewis (2020). Increasing trends in regional heatwaves. Nature communications, 11(1), 3357. At least a mention to their studies would improve the introduction section related to this extreme events.
3. In relation to wildfires, I also recommend some more references, such as Turco et al., (2018). Exacerbated fires in Mediterranean Europe due to anthropogenic warming projected with non-stationary climate-fire models. Nature communications, 9(1), 3821, or Jones et al. (2022). Global and regional trends and drivers of fire under climate change. Reviews of Geophysics, 60(3), e2020RG000726.

Our response

We thank Reviewer #2 for the literature suggestions and added them to the introduction but found them to be more appropriate in line 63.

Reviewer #2

4. Lines around 85: I am not particularly familiar with ISIMIP project, so I am not sure how many impactextreme events could be taken from there, but perhaps a comment or indication of why these three events are considered instead of others, and if some others could also be used to apply this methodology in a similar way.

Our response

We thank Reviewer #2 for pointing out this missing explanation. In principle, ISIMIP also allows to define extreme events within the hydrological sector while other sectors are still work in progress (<https://www.isimip.org/about/#sectors-and-contacts>). In the case of hydrological event we are still working on additional impact model simulations and may need a different extreme event setup since extreme flood events are typically derived from additional steps, e.g. calculation and distribution of discharge.

We clarified these aspects in the Introduction.

Changes in the manuscript:

In Introduction:

“Note that also hydrological models take part in ISIMIP which would allow to define hydrological extreme events but due to the intricacies of the related model setups we restrict ourselves to agricultural and vegetation models and use heatwaves as a direct GCM derived event.”

Reviewer #2

5. Line 99: When the *dominant period* method is defined, it seems that it is a novel approach, and so no previous references to other studies can be made, even applied to other areas of research. If it is the case, perhaps a more clear statement about its novelty should be made, for a reader to know that it has not been used before.

Our response

Thank you. Indeed, the definition of the dominant period is a novel approach and we clarified it accordingly.

Changes in the manuscript:

In Introduction:

“This novel approach combines Fourier analysis with statistical tools and is motivated by the large size of the spatio-temporal data set within our multi-model setup, which calls for a simple characterization of regularity.”

Reviewer #2

6. Line 138: How long does it last the pre-industrial control period, or how are they defined such conditions?. It seems that in reference 34, the web page of ISIMIP project, could be found, but it is not very clearly accessible, and some information should be given to understand this reference conditions. Ok, when arriving to section 4, I understand that

my concerns are described there, but it makes not much sense to me such manuscript structure. It seems that section 4 should be made section 2, just after introduction, to ease the understanding of results, to my opinion.

Our response

We thank Reviewer #2 for pointing out this missing connection. The manuscript structure is a journal requirement but we added a reference to Sec. 4 and clarified the period length at the beginning of Section 2.

Changes in the manuscript:

In Results:

“The pre-industrial control (picontrol) setup within the ISIMIP framework [ISIMIP3] simulates stable pre-industrial climate conditions from 1850 to 2100 and serves as a reference for the future SSP scenarios (see Sec. 4).” (underlined sections are new)

Reviewer #2

7. Let me go then to section 4, and then back to sections 2 and 3. On line 405, I again fully suggest that Perkins reference studies should be mentioned, at least the review one: Perkins (2015). A review on the scientific understanding of heatwaves—Their measurement, driving mechanisms, and changes at the global scale. Atmos. Res., 164, 242-267, for a more robust literature context, despite the HWMId index is more deeply described, as it is in the Russo et al. papers.

Our response

We thank Reviewer #2 for the suggestions and added the reference accordingly.

Reviewer #2

8. Line 418. 50 years is chosen, based on some sensitivity tests, if I am right. But what about using 30 years, as it seems to be the usual time length chosen by WMO to define a climatology?

Our response

The length of the time window was chosen to encompass the relevant climate modes that were observed to our extreme event indicators, such as ENSO, NOA, and IOD. Indeed, we show a test with window length $\Delta T=30y$ which corresponds to a total length of $2\Delta T=60y$ since the auto correlation function is based on a fixed window and a moving window (see Sec. 4 step 1 and Supplementary Material Fig. S2). The results do not change qualitatively except for a reduced signal due to the decorrelation in a longer time window.

We clarified this aspect.

Changes in the manuscript:

In Methods:

“By increasing the window size to $2\Delta T = 60y$ we find no qualitative difference except for the reduced signal due to stronger decorrelation within the longer time window (see Supplementary Discussion Sec. 1.1).”

Reviewer #2

9. Now going back to results section (lines after 117), I wonder if first a figure of how crop failure, heatwaves and wildfires spatial averaged distribution themselves, for example, averaged over the pre-industrial control period would look like would ease the understanding of the proposed method to describe how they could change their regularity due to global warming. Each of these phenomena are quite different in their overall structure, so their changes could be largely dependent on their pre-industrial climate description, and for a non expert reader on some of these extreme events, it could be hard to understand what do the change would mean, and also for a better understanding of the comparison among the three types of events.

Our response

We thank Reviewer #2 for the suggestion and added an additional panel of globally aggregated affected area in Fig. 2 based on picontrol to improve the comparison between the picontrol scenario and SSP5-8.5. Due to the drastic increase in events within SSP5-8.5 compared to picontrol extreme event time series will be highly affected by these changes. These changes are indeed different for each event category and we therefore observe different changes in the dominant periods.

Changes in the manuscript:

In Results:

“The stability of climatic pre-industrial conditions results in low variance of total affected area for all three event categories (see Fig. 2 d-f).”

Reviewer #2

10. Cropland failure: mixing all crops, both irrigated and rainfed conditions to obtain average return periods (lines 133-138) could make the interpretation of this index more complex to be made?. Would it be more clear if a single crop, the more extended one, or made it separately, the numbers or results be quite different from the ones shown on that figure 1.a?

Our response

We thank Reviewer #2 for the suggestion. We ran the calculation for the aggregated crop yield to account for the specific land use pattern at each grid cell but indeed crop types are affected differently across irrigation and crop type (Heino2018, Anderson2019) which leads to different time series and therefore different potential dominant periods.

In order to estimate the effect of this aggregation we analysed dominant periods for the largest crop type at each grid cell and show the results in Supplementary Materials Sec. 5. We indeed observe differences in the occurrence of dominant periods while the dominant periods (7-13 years) are consistent with the reported results in the main manuscript.

Changes in the manuscript:

In Results:

“Calculating dominant periods for each crop type separately leads to the same dominant periods but with a different count and spatial distribution (see Supplementary Fig. 7).”

In Supplementary Material:

“Crop resolved dominant period

Since crop yields are differently affected by climate modes [Heino2018, Anderson2019, Heino2020] we calculate dominant periods for each crop type separately.

Figure 7: Crop resolved dominant periods. Median dominant period for (a) maize, (b) rice, (c) soy, and (d) wheat crop failure for picontrol aggregated over all time windows 1850-1899, 1900-1949, ..., 2050-2099 and climate-impact models. The white color signifies no extreme climate impact occurrence and gray color signifies no dominant period (irregularity) while existing dominant periods are grouped in three-year regularity intervals ranging from 1-4 years (red) to 22-25 years (black).

We find the same dominant periods across all crop types (see Fig. 7) and consistent with the totally aggregated result in the main text Fig. 1 a.

The main difference between crop type is the regional distribution of dominant periods. For example, dominant periods for maize crop failure occur on all continents while for rice we have a concentration in South America and South/South East Asia in agreement with [Heino2020].

In the case of soy we find a dominant periods mainly in the Americas whereas yield fluctuation influences from ENSO are reported in more world regions [Heino2020].

Similarly, we find dominant periods across all world regions while reported influences are fewer in Europe and Asia [Heino2020].”

Reviewer #2

11. Section 2.1.2. Heatwaves. I am not sure to understand such result, to be sincere. How it can be understood such “absence” of regularity or dominant period for heatwaves anywhere for the preindustrial period?. In line 177 it is related to the natural variability, but I feel some more explanation should be given, as it seems somewhat counter intuitive, or maybe that the proposed index of regularity of extreme heat conditions.

Our response

The absence of dominant period is due to the extreme definition of heatwaves which is (now) characterized by the HWMI_d exceeding the 97.5th percentile of the picontrol distribution. In Fig. 1 we consider picontrol which means that the condition on HWMI_d leads to an extreme event time series where on average 97.5% of the time steps show no extreme event (0) and only 2.5% of the time steps show an extreme event (1). Consequently our 50-years windows show on average only $2.5\% * 50 = 1.25$ events. This is also visible in Fig. 6 (b) in the time interval 1850-1899 where also only one event is present. With only 1.25 events on average there can be no dominant period because one event is not sufficient to define any regularity.

We further clarified this aspect in the manuscript (see response to Reviewer #1 above), thank you.

Reviewer #2

12. In figure 1, perhaps the total distribution box should have the same vertical axis, from 0 to 1 for the three impacts. I have a small concern about the fact that such number should be given in percentage, and how to deal with the huge difference in white (no occurrence) areas among the three indices, to have a more clear homogenization among them on that figure.

Our response

We thank Reviewer #2 for the suggestion and agree that the comparability between events needs to be improved. First, we adjusted the definition of heatwaves by dropping the second condition for heatwaves, namely that the Humidex needs to exceed 45 at all hot days defined through the HWMI_d, to have similar affected areas under piconrol (see also responses to Reviewer #1). Additionally, we adjusted the y-scale for all three impact types as suggested, namely ranging from 0 to 1.

Reviewer #2

13. About ENSO role on the changes obtained for climate change scenarios on each of the extreme impactsevents, I would be more careful, as, although references are indicated that could explain or relate this climate variability pattern to wildfires or crop failures, it is probably much more complex that those potential correlations, and for sure it is impossible to affect the same way to the three extreme events used in this work. I mean, for example, on line 301, when talking about heatwaves.

Our response

We agree that the relationship between climate modes and extreme events is very complex and not direct. Therefore, we have extended our analysis by calculating the correlation between affected areas and ENSO (see response to Reviewer #1) but also extended the literature review in the Introduction (see response to Reviewer #1). The correlation for heatwaves 0.37 (Supplementary Material Sec. 9) which shows that there is a relationship but not a strong determining one.

In addition to the changes mentioned above we clarified this aspect in line 301:

Changes in the manuscript:

In Results:

“This increase towards smaller dominant periods may be not only related to the general increase in temperature within SSP5-8.5 but also to influences from stronger ENSO effects with a fingerprint in the 2 to 7 years range.” (underlined insertion is new)

Reviewer #2

14. Detrended computations are quite relevant on section 2.2 analysis. I find it quite interesting suchmathematical procedure. But at the same time, this linear detrended analysis can be tricky when applied to different indices, as it is the case here. Maybe a more precise explanation than what is said on line 252, that equation 2 is just briefly mentioned, could be indicated.

Our response

We thank Reviewer #2 for the suggestion. The procedure consists of an ordinary least squares (OLS) regression for each univariate correlation function. From the OLS regression we take the slope component subtract this one from the correlation function. We added a more explicit description.

Changes in the manuscript:

In Results:

“Linear detrending consists of an ordinary least squares regression for each univariate correlation function in each time window (see Eq. (2)) and subtracting the slope component from the time correlation function.”

Reviewer #2

15. I have some comments about heatwaves analysis. Figures just show land heatwaves, but why not including also marine heatwaves, that have become a relevant area of research on the past years?, see for example Oliver et al. (2021). Marine heatwaves. Annual review of marine science, 13(1), 313-342 or Capotondi et al. (2024). A global overview of marine heatwaves in a changing climate. Communications Earth & Environment, 5(1), 701. Maybe ISIMIP does not include these regions on their database, but let me make such comment about its importance. I also had the feeling that perhaps a more consistent and direct comparison of the proposed method could be seen if also coldwaves were analyzed. How do the authors feel about it, and other climate impact events extension of the proposed study?.

Our response

We thank Reviewer #2 for pointing out different types of temperature-related events. Indeed, the methodology is applicable to all types of events but the main limitation is data availability. Within ISIMIP marine temperature data is not (yet) available but we extended the manuscript on possible other temperature-related extreme event definitions.

Our goal in this manuscript is to present the new measure for dominant periods and investigate the results for three different impact categories. Including more/different temperature related extreme event is a very interesting question requiring a broader discussion on different heat-/coldwave indices. While we are definitively interested in this direction we believe that it goes beyond the scope of this study.

Changes in the manuscript:

In Methods:

“Note that the impacts of heatwaves depend on the applied definition and that heatwave definitions may also be based on marine systems [Perkins2015, Oliver2021].”

Reviewer #2

16. Two final remarks. The 50 year time windows periods chosen for the analysis, and in particular (2051-2100) for the future climate change period can make sense, but I guess if the more classical 30 year periods during the XXIst century (2041-2070, 2071-2100) could add more interest to the study, being also statistically robust enough for the proposed analysis.

Our response

We thank Reviewer #2 for the suggestion and added the time windows 2040-2069 (main text) and 2070-2099 (Supplementary Material) to our analysis.

While both periods are similar to the extended 2050-2099 period results in terms of dominant periods and the shift towards higher frequency, we find a significant difference for heatwaves at the end of the century. This difference is due to strong non-linear warming trend which requires a smaller time window (2070-2099) to compensate to absorb the warming trend sufficiently and recover dominant periods. Apart from this detrending effect we find a similar distribution of dominant periods.

Changes in the manuscript:

In Results:

“This transition becomes more evident in ~~2050-2100~~ 2040-2069 (Fig. 3 b) where ~~almost all~~ most world regions show only long dominant periods or no regularity. Note that we are considering a smaller time window at the end of the century to avoid the strong monotonic warming trend in extreme events which mainly determines the results in 2050-2099 (see Supplementary Discussion Sec. 9).

The largest detectable dominant period in the time window 2040-2069 is 15 years but the largest dominant periods of 22-25 years are observed when the time window 2050-2099 is considered (see Supplementary Figure 29 a).”

“The corresponding median dominant periods for ~~2050-2100~~ 2040-2069 are shown in Fig. 3 c. Removing the trend in the correlation function leads to a relative increase in irregularity and to a relative decline in largest dominant periods, which is consistent with the assumption that large dominant periods result from the global warming trend that can be ~~partially~~ mostly absorbed through detrending.”

“~~These signals are still few compared to the overall majority of dominant periods between 22 and 25 years. We attribute this small change to the superlinear increase in affected area (see Fig. 2 (a)) which cannot be fully absorbed through linear detrending. Note that by considering the $2\Delta T=50y$ we find more irregularity in the time window 2050-2099 due to the superlinear warming effect that cannot be sufficiently absorbed with linear detrending (see Supplementary Fig. 29)~~”

In Methods:

“In the strong warming scenario we additionally consider $2\Delta T=30$ years to account for the strong warming trend.”

In Supplementary Material:

Section 9 (due to the large number of figures we refrain to post the full section here)

Reviewer #2

17. Due to the global covering of the proposed analysis, and taking into account the quite large spatial heterogeneity's of such climate impact events, did the authors thought about doing some subregional analysis, at least, using some of the main regions defined at IPCC reports?.

Our response

We thank Reviewer #2 for the suggestion. Indeed, we considered doing a regionally aggregated analysis but since extreme events typically occur on smaller time and spatial scales did not pursue this idea. On the other hand, spatial aggregation may be interesting to see how spatial fluctuations (as mentioned by Reviewer #2) may cancel out through aggregation and therefore further stabilize the results. We therefore calculated these regionally aggregated extreme event time series and computed the corresponding dominant periods.

We aggregated (area weighted average) the time series to the following regions: North America (NAC), East Asia & Pacific (EAS), Europe & Central Asia (ECS), Latin America & Caribbean (LCN), Middle East & North Africa (MEA), South Asia (SAS),

and Sub-Saharan Africa (SSF). Due to the regional differences we observe a cancellation of spectral features and phases and observe no dominant periods. For example, we find the following regional correlation function in the case of wildfire.

REVIEWER COMMENTS

Reviewer #1

The authors have done an impressive job in addressing all my technical questions and conducted many additional analyses and partly revised their results.

Our response

We thank Reviewer #1 for the positive assessment of our revisions.

Reviewer #1

The only open question now is the one on relevance: I am still not convinced about the concept of dominant periods, I did not find the arguments convincing:

(1) The argument that a classical AR framework would be computationally much more demanding (because of the model selection) is not quite correct. The classical framework intrinsically provides a theory with associated uncertainty estimates and information on the "sharpness" of the dominant frequency. Obtaining the same information from the proposed new framework would require substantial bootstrapping/Monte Carlo simulations and likely lead to the same (or even more) computational cost. This holds in particular as the model selection could be limited to 2nd order models, as only dominant periods are sought.

Our response

We thank the reviewer for pointing out this aspect.

Our argument on the computational cost is based on the model selection, i.e. which AR(n) model to choose for a given time series. If m is the maximum lag we want to consider, there are 2^m possible AR(m) models that need to be fitted to our time series and compared to each other. In our case we are typically dealing with time windows of 25 years which means that even for a maximum lag of 10 we have to compare $2^{10}=1024$ AR models for each grid cell, scenario, model combination, time window, and extreme event type. Consequently, finding the optimal AR model within our multi-model setup poses computational limitations.

We agree that AR models are embedded in a broad mathematical theory that allows to estimate the statistics of spectral peaks. In our case where each climate impact model produces single realizations we could use extensions to classical Fourier analysis such as the Multi Taper method to obtain uncertainty estimates (Thomson 1982, Percival and Walden 1993). Unfortunately, this extension would introduce new degrees of freedom (e.g. number of tapers, choice of tapers) and would come at the cost of spectral resolution. To estimate uncertainties of dominant periods we calculate the standard deviation across all models and time windows (see Sec. 4.3). In the case of the picontrol runs the standard deviation is in most cases smaller or equal 1 year (Supplementary Fig. 33).

Regarding the model selection we have checked whether the optimal AR(n) model (based on AIC) leads to the same strongest period as the AR(2) model:

	Dominant period (Fourier analysis)	Strongest period (Optimal AR(n) model)	Lags of optimal model	Strongest period from AR(2) model
Example 1 (Fig. 7 (a))	8.33y	15.53y	1, 2, 4, 5, 6, 7, 9, 10, 11	2y
Example 2 (Fig. 7 (b))	None (12.5y candidate)	6.88y	2, 3, 4, 6, 9, 10, 11	6.1y
Example 3 (Fig. 7 (c))	25y	7.63y	1, 3, 4, 5, 6, 8, 10	None (spectral peak zero frequency)

We find that the optimal AR(n) model differs significantly from the AR(2) model as it contains several additional coefficients from lags larger than 2. Also, the models yield different strongest spectral peaks.

We clarified our approach of determining the optimal AR(n) model in the new version of the manuscript.

Changes in the manuscript:

In Sec. 11 Supplementary Discussion:

Replace Supplementary Table 2 by

	Dominant period (Fourier analysis)	Strongest period (Optimal AR(n) model)	Lags of the optimal AR(n) model
Example 1 (Fig. 7 (a))	8.33y	15.53y	1, 2, 4, 5, 6, 7, 9, 10, 11
Example 2 (Fig. 7 (b))	None (12.5y candidate)	6.88y	2, 3, 4, 6, 9, 10, 11
Example 3 (Fig. 7 (c))	25y	7.63y	1, 3, 4, 5, 6, 8, 10

“In addition, we estimate the optimal set of lag parameters for the autoregressive model through the AIC information criterion by comparing all possible model setups as implemented in `ar_select_order` method of the AR class in statsmodels [14]. We restrict the maximum lag of possible AR(n) models to 11 due to the exponential growth of possible model combinations.” (underlined sentence is new)

Reviewer #1

(2) Also the additional information compared to return periods is questionable. It would only, if it would allow for a more precise estimation of waiting times between events, but arguments for such a potential benefit are not provided by the authors. "Dominant period" does not mean that the information is strictly periodic, so no precise waiting time until an event can be derived. And given the lack of a statistical theory, prediction intervals for the waiting time cannot easily be calculated. Again, Monte Carlo simulations would be required where a classical AR2 framework might give "cheaper" information.

As the focus of the paper is still on the new concept, I see it more in a technical journal, where new analysis methods can be presented.

Our response

We agree with Reviewer #1 that dominant periods do not imply strict periodicity. The added value of the dominant period is that this single period and its higher harmonics can explain at least half of the time series variability as implemented through the R^2 criterion. Consequently, it provides information on the regularity of the underlying noisy time series.

This concept is also different to statistical waiting/recurrence times which estimate the time passing between two extreme events. Our example in the Introduction highlights this difference in terms of the extreme event time series (1,0,1,0,1,0,1,0) and (1,1,0,0,0,1,0,1). Both time series have recurrence times 2 years but the additional information that the first time series has a dominant period of 2 years while the second one has no dominant period provides valuable information to differentiate between both time series.

We have further clarified the error estimation in the new version of the manuscript.

Changes in the manuscript:

In Section 4.3:

“An estimate for the difference in dominant periods between different climate impact model combinations and time windows is given in terms of the standard deviation (see Supplementary Discussion Sec. 12). Note that the error estimation is limited due to the single realizations from the climate impact models. The standard deviation for all event categories is concentrated in the small range of 0 to 4 years.” (underlined sentence is new).

Reviewer #2

Reviewer #2 (Remarks to the Author):

None

Reviewer #2 (Remarks on code availability):

None

Review for Nature Communications NCOMMS-24-85559

Title: “Shifting (ir)regularity regimes in extreme climate impacts under global warming”

Authors: Karim Zantout et al.

I find the study quite interesting for publication. I hope that my requests, comments and concerns can be answered satisfactorily by the authors, and under the consideration of the editorial board of the journal.

1. General comments

1. Abstract: “The”, as a generic expression of extreme climate events seems to be confusing and potentially leading to misunderstanding when reading the abstract. I would suggest “Several”, “Plenty”, “Many”, or perhaps nothing, and so state directly “Spatio-temporal patterns”. And of course, state that the work is going to focus on three examples or specific events: crop failures, heatwaves and wildfires. Otherwise, it could be thought that it can be applied/generalized to other, among many, extreme events: droughts or dry spells, heavy rains, wind extremes, coldwaves, etc..
2. Line 65: Although it is not fully related to heatwaves temporal regularity patterns, the work of Molina et al., (2020), <https://doi.org/10.1038/s41598-020-65663-0>, figures 4-5 show time trends over several subregions over Europe, and so it could be named on that list of references, and also on the comment at line 81. When mentioning heatwaves characterization, I strongly suggest also some of the many works led by Sarah Perkins, known as one of the main researchers on these extreme events. See for example, on this same journal, Perkins-Kirkpatrick and Lewis (2020). Increasing trends in regional heatwaves. Nature communications, 11(1), 3357. At least a mention to their studies would improve the introduction section related to this extreme events.
3. In relation to wildfires, I also recommend some more references, such as Turco et al., (2018). Exacerbated fires in Mediterranean Europe due to anthropogenic warming projected with non-stationary climate-fire models. Nature communications, 9(1), 3821, or Jones et al. (2022). Global and regional trends and drivers of fire under climate change. Reviews of Geophysics, 60(3), e2020RG000726.
4. Lines around 85: I am not particularly familiar with ISIMIP project, so I am not sure how many impact extreme events could be taken from there, but perhaps a comment or indication of why these three events are considered instead of others, and if some others could also be used to apply this methodology in a similar way.
5. Line 99: When the *dominant period* method is defined, it seems that it is a novel approach, and so no previous references to other studies can be made, even applied to other areas of research. If it is the case, perhaps a more clear statement about its novelty should be made, for a reader to know that it has not been used before.
6. Line 138: How long does it last the pre-industrial control period, or how are they defined such conditions?. It seems that in reference 34, the web page of ISIMIP project, could be found, but it is not very clearly accessible, and some information should be given to understand this reference conditions. Ok, when arriving to section 4, I understand that my concerns are described there, but it makes not much sense to me such manuscript structure. It seems that section 4 should be made section 2, just after introduction, to ease the understanding of results, to my opinion.
7. Let me go then to section 4, and then back to sections 2 and 3. On line 405, I again fully suggest that Perkins reference studies should be mentioned, at least the review one: Perkins (2015). A review on the scientific understanding of heatwaves—Their measurement, driving mechanisms, and changes at the global scale. Atmos. Res., 164, 242-267, for a more robust literature context, despite the HWMId index is more deeply described, as it is in the Russo et al. papers.
8. Line 418. 50 years is chosen, based on some sensitivity tests, if I am right. But what about using 30 years, as it seems to be the usual time length chosen by WMO to define a climatology?
9. Now going back to results section (lines after 117), I wonder if first a figure of how crop failure, heatwaves and wildfires spatial averaged distribution themselves, for example, averaged over the pre-industrial control period would look like would ease the understanding of the proposed method to describe how they could change their regularity due to global warming. Each of these phenomena are quite different in their overall structure, so their changes could be largely dependent on their pre-industrial climate

description, and for a non expert reader on some of these extreme events, it could be hard to understand what do the change would mean, and also for a better understanding of the comparison among the three types of events.

10. Cropland failure: mixing all crops, both irrigated and rainfed conditions to obtain average return periods (lines 133-138) could make the interpretation of this index more complex to be made?. Would it be more clear if a single crop, the more extended one, or made it separately, the numbers or results be quite different from the ones shown on that figure 1.a?
11. Section 2.1.2. Heatwaves. I am not sure to understand such result, to be sincere. How it can be understood such “absence” of regularity or dominant period for heatwaves anywhere for the pre-industrial period?. In line 177 it is related to the natural variability, but I feel some more explanation should be given, as it seems somewhat counter intuitive, or maybe that the proposed index of regularity of extreme heat conditions.
12. In figure 1, perhaps the total distribution box should have the same vertical axis, from 0 to 1 for the three impacts. I have a small concern about the fact that such number should be given in percentage, and how to deal with the huge difference in white (no occurrence) areas among the three indices, to have a more clear homogenization among them on that figure.
13. About ENSO role on the changes obtained for climate change scenarios on each of the extreme impacts events, I would be more careful, as, although references are indicated that could explain or relate this climate variability pattern to wildfires or crop failures, it is probably much more complex that those potential correlations, and for sure it is impossible to affect the same way to the three extreme events used in this work. I mean, for example, on line 301, when talking about heatwaves.
14. Detrended computations are quite relevant on section 2.2 analysis. I find it quite interesting such mathematical procedure. But at the same time, this linear detrended analysis can be tricky when applied to different indices, as it is the case here. Maybe a more precise explanation than what is said on line 252, that equation 2 is just briefly mentioned, could be indicated.
15. I have some comments about heatwaves analysis. Figures just show land heatwaves, but why not including also marine heatwaves, that have become a relevant area of research on the past years?, see for example Oliver et al. (2021). Marine heatwaves. *Annual review of marine science*, 13(1), 313-342 or Capotondi et al. (2024). A global overview of marine heatwaves in a changing climate. *Communications Earth & Environment*, 5(1), 701. Maybe ISIMIP does not include these regions on their database, but let me make such comment about its importance. I also had the feeling that perhaps a more consistent and direct comparison of the proposed method could be seen if also coldwaves were analyzed. How do the authors feel about it, and other climate impact events extension of the proposed study?.
16. Two final remarks. The 50 year time windows periods chosen for the analysis, and in particular (2051-2100) for the future climate change period can make sense, but I guess if the more classical 30 year periods during the XXIst century (2041-2070, 2071-2100) could add more interest to the study, being also statistically robust enough for the proposed analysis.
17. Due to the global covering of the proposed analysis, and taking into account the quite large spatial heterogeneity's of such climate impact events, did the authors thought about doing some subregional analysis, at least, using some of the main regions defined at IPCC reports?.